# Modelling Long-Term Industry Energy Demand and CO$_2$ Emissions in the System Context Using REMIND (Version 3.1.0)

Michaja Pehl[1], Felix Schreyer[1], and Gunnar Luderer[1]

[1]Potsdam Institute of Climate Impact Research, PO Box 60 12 03 Potsdam, Germany

**Correspondence:** Michaja Pehl (michaja.pehl@pik-potsdam.de)

**Abstract.** This paper presents the extension of industry modelling within the REMIND integrated assessment model to industry subsectors, and the projection of future industry subsector activity and energy demand for different baseline scenarios for use with the REMIND model. The industry sector is the largest greenhouse gas-emitting energy demand sector and considered a mitigation bottleneck. At the same time, industry subsectors are heterogeneous and face distinct emission mitigation challenges. By extending the multi-region, general equilibrium integrated assessment model REMIND to an explicit representation of four industry subsectors (cement, chemicals, steel, and other industry production), along with subsector-specific carbon capture and sequestration (CCS), we are able to investigate industry emission mitigation strategies in the context of the entire energy-economy-climate system, covering mitigation options ranging from reduced demand for industrial goods, over fuel switching and electrification, to endogenous energy efficiency increases and carbon capture. We also present the derivation of both activity and final energy demand trajectories for the industry subsectors for the use with the REMIND model in baseline scenarios, based on short-term the continuation of historic trends and long-term global convergence. The system allows for selective variation of specific subsector activity and final energy demand across scenarios and regions to create consistent scenarios for a wide range of socioeconomic drivers and scenario story lines, like the shared socioeconomic pathways (SSPs).

## 1 Introduction

The targets of the Paris Accord (United Nations, 2015) (holding global warming to well below 2°C while pursuing efforts to limit it to below 1.5°C) have been shown to entail limited budgets of future CO$_2$ emissions (Matthews and Caldeira, 2008, Meinshausen et al. (2009), Pathak et al. (2022)). The CO$_2$-budgets for reaching the 1.5°C limit with a likelihood of 50 % was estimated at around 500 GtCO$_2$ by the latest IPCC assessment report (Arias et al., 2021). Similarly, limiting warming to below 2°C with 67 % likelihood – a possible interpretation of the well below 2°C target (United Nations Environment Programme, 2022) – is assessed to constrain future cumulative CO$_2$ emissions to 1150 GtCO$_2$. While these estimates are highly uncertain in size (Rogelj et al., 2019), it is evident that they are so constrained as to be unattainable without carbon dioxide removal (CDR) (Luderer et al., 2018). The availability of CDR, however, is uncertain (Smith et al., 2015, Fuss et al. (2018), Minx et al. (2018)) and subject to major sustainability concerns (Smith et al., 2015). Therefore, minimizing residual fossil CO$_2$ emissions from the energy supply, buildings, industry, and transport sectors is imperative to limit reliance on CDR.

In 2020, industry accounted for 26 % of global $CO_2$ emissions from fuel combustion and processes (International Energy Agency, 2021) and is the largest emitting demand sector. Industry is broadly considered a decarbonisation bottleneck. Low carbon solutions exist for almost all industrial processes, but decarbonisation is stymied by high abatement costs, inertia caused by the longevity of installations and the technology development, as well as limited availability of (low-cost) low-carbon energy inputs (Bashmakov et al., 2022, Deason et al. (2018)). While technological options for electrifying most of industrial use of

fuels are available, or can be in the foreseeable future (Madeddu et al., 2020), their implementation depends on reductions of high electricity prices (relative to fossil fuels) (Luderer et al., 2022). Widespread electrification of industry, together with rapid decarbonisation of power supply, would alleviate most of the combustion emissions of the sector, leaving process emissions (e.g. reduction of iron ore, calcination of limestone), which need to be addressed by either negative emissions or modified process designs. A major driver of industry activity and emissions is the demand for industrial goods and services, which is

underpinning both economic growth and increasing consumption levels, and demand reductions are discussed as a climate mitigation option (Bashmakov et al., 2022).

In view of the importance of the industry sector in terms of energy demand and $CO_2$ emissions, its accurate modelling in the context of the overall energy transition is crucial for deriving meaningful climate change mitigation pathways in line with the Paris climate targets. Also, the industry sector competes with other sectors – buildings, transportation or energy

supply – for low-carbon energy, a dwindling carbon budget as well as finite geological capacity for carbon sequestration. The transformation of industry therefore needs to be analysed in the context of these other systems (energy production, other energy demand sectors, and $CO_2$ management options). Integrated Assessment Models (IAMs) provide a combined perspective on the integrated energy-economy-climate system.

So far, IAMs either pursue a top-down approach with strong limitations in the granularity of industry subsector and process

representation, or stronger bottom-up representation of processes at the expense of an incomplete representation of system integration, e.g., in terms of macroeconomic substitutions and implications for material demand the impact of climate policies on economic growth.

We here present a newly developed industry sector module for the REMIND model which seeks to represent characteristics of processes in key subsectors while at the same time fully embedding the industry sector in the modelling of the transformation

of the energy-economic system. Specifically, the new module explicitly represents the industry subsectors cement, chemicals, steel production (split into primary and secondary production), as well as combined other industries. It models CCS individually per subsector, switching to advanced final energy carriers (hydrogen and electrification of high-temperature heat provision), energy efficiency investments, and links the industry sector to both macroeconomic demand drivers as well as final energy provision and carbon management of a detailed bottom-up energy system model. It thus covers all major mitigation options

for industry, ranging from end-of-pipe approaches to fuel switching, electrification and efficiency as well as macroeconomic adjustments in the response to climate policy.

There is a lack of integrated assessment models representing price and economic feedbacks on industrial demand as well as efficiency, fuel switching and CCS options in industrial activities. In this respect, the extended industry sector in the REMIND model compares favourably to other IAMs. A forthcoming overview of industry $CO_2$ emission reduction (Bauer et al., in

preparation) compares seven IAMs "with improved industry sector representation". They all represent at least the cement, chemicals, and steel production subsectors, and to varying degrees mitigation options like energy efficiency improvements, final energy substitution, and CCS. The endogenous reduction of industry demand as a trade-off with other mitigation options, is a feature of general equilibrium models (two out of the seven: GEM-E3 (Fragkos and Fragkiadakis, 2022) and REMIND). Partial equilibrium models ((COFFEE: Rochedo (2016), MESSAGEix: Grubler et al. (2018), IMAGE: van Sluisveld et al. (2021), POLES: Després et al. (2018), PROMETHEUS: Fragkos et al. (2015)) rely on exogenous demand pathways for policy scenarios and are typically inelastic to changes in $CO_2$ and thus energy prices. The WITCH model (The WITCH team, 2017) is an example of another general equilibrium IAM, but only represents an aggregated stationary and not a separate industry sector.

Further to the refinements of industry sector representation within REMIND, we present new methods for producing consistent trajectories for industry subsector production and final energy use for different baseline scenarios used for calibrating the REMIND model. They project per-capita subsector activity based on per-capita GDP projections and allow for the variation of material and energy intensity to derive different scenarios.

## 2 The industrial production system in the macroeconomic context

### 2.1 REMIND Model Structure

The integrated assessment model REMIND (Baumstark et al., 2021) is comprised of a macroeconomic growth model and a detailed energy system model (ESM), which are hard-linked, i.e. optimised simultaneously, with energy supply (by the ESM) and demand (by the macroeconomic model) quantities and prices in equilibrium. The ESM represents over 50 conventional and low-carbon energy conversion technologies, modelling energy flow from primary through secondary to final energy, accounting for $CO_2$ and other greenhouse gas (GHG) emissions as well as carbon capture and sequestration (CCS), carbon capture and utilisation (CCU) and options for carbon dioxide removal (CDR). The macroeconomy is represented by an intertemporal general equilibrium model that maximises intertemporal welfare of twelve to 21 world regions (depending on parametrisation) that are linked by trade in primary energy carriers, an aggregated trade good, and emission permits. The macroeconomic model uses a nested constant elasticity of substitution (CES) production function (Jae Wan Chung, 1994), in which labour, energy, and capital are used for production of economic output, and energy inputs are tracked through the economic sectors buildings, industry, and transport to final energy carriers which constitute the links to the ESM. The different economic sectors form themselves sub-trees in the CES function and can be realised in different levels of detail. This paper describes specifically the extension of the industry sector from an aggregated realisation to one in which different subsectors are modelled explicitly. The nested CES production function takes the form

$$V_o = \left( \sum_{(o,i) \in \text{CES}} \alpha_i V_i^{\rho} \right)^{\frac{1}{\rho}} \tag{1}$$

where $V_o$[1] is the output quantity, $V_i$ are the input quantities, $\alpha_i$ are efficiency parameters for the inputs, and $\rho_o$ is a parameter derived from the substitution elasticity $\sigma_o$ ($\rho = 1 - \sigma^{-1}$) between the inputs on a CES nest. The set CES of tuples $(o, i)$ links the output of a CES nest to its inputs. Since outputs on one level are inputs on another, this gives rise to a tree structure (cf. Figure 1). Substitution elasticities $\sigma$ in general describe the relative change in utilisation of the inputs to an economic production process (called production factors) in relation to the relative change in input prices. A constant elasticity of substitution function assumes that a specific percentage increase (decrease) in the price of one production factor in relation to the prices of the other factors will cause a constant specific percentage decrease (increase) in factor utilisation, irrespective of the amount of factor utilisation at which this price change occurs.

Operating the REMIND model entails calculating a baseline scenario, in which no climate change mitigation policies are assumed, and then imposing different constraints on the model (e.g. a fixed carbon price trajectory, a limit on peak and/or end-of-century temperature increase, a limited greenhouse gas emissions budget over the century) for calculating policy scenarios that are used to investigate specific research questions (e.g. the feasibility of climate change mitigation targets under constrained availability of technologies, the impact of climate change mitigation policy on investments into certain technologies, the macroeconomic costs of different climate change mitigation regimes) (Baumstark et al., 2021). Besides initialising a large set of variables (i.a. all stocks of energy conversion technologies in the ESM, their efficiency parameters, trade in all traded goods) for the first model time step (2005), this requires determining the efficiency parameters $\alpha_i$ for the CES production function for the entire model time horizon. For this, trajectories for the output (macroeconomic production – GDP) of the CES production function and all inputs (final energy demand and for industry also subsector production levels) into the CES production function over the entire model time horizon (until 2100) are needed.

In addition to extending the industry sector in the REMIND model itself, it is also necessary to calculate consistent trajectories for the input data of the CES calibration (subsector production and final energy demand) for different baseline scenarios of future development (see section 3.1).

## 2.2 Industry Subsectors

The CES production function of REMIND can be expanded to explicitly represent the industrial subsectors and economic drivers for the demand of their products. The industry sector of the REMIND model is split into four subsectors: cement[2], chemicals, steel production, and all "other industry" production, based on the energy demand characteristics of the sectors, their portion in industry energy demand and $CO_2$ emissions, and the applicability of CCS.

The products of both the cement and steel subsectors are comparatively homogeneous with well-defined properties, for which country-level statistics in physical units are available, and the subsectors are dominated by few production processes. The chemicals subsector produces a diverse range of intermediate and final products, yet some energy intensive processes (e.g. steam cracking or reforming) are common to many production routes (Fischedick et al., 2014). Since production statistics

---

[1]Most variables in REMIND vary with time. We omit time indices unless they are relevant for the equation at hand.

[2]Due to limited data availability, the "cement" subsector in the REMIND model encompasses the entire "non-metallic minerals" subsector, of which cement production is the dominant part.

for the different products are not widely available and the products lack commensurability with respect to energy inputs and $CO_2$ emissions of production, a monetary measure is employed to model the activity of the chemicals subsector. The last subsector, "other industry", is by definition characterised by diverse processes and heterogeneous goods which are not comparable on a physical basis, and therefore a monetary measure is used for modelling its activity, too. The decarbonisation challenges faced by "other industry" production are, however, quite comparable, as most of the energy demand can be electrified by established technologies (Madeddu et al., 2020).

Using value added instead of physical production to drive industry energy demand incurs two difficulties. Both the specific value added per unit of (physical) production and the composition of different types of products making up subsector production vary across regions and change over time. This reduces the interpretability of subsector production figures given in value added, especially in absolute terms. It does, however, not impinge on the usefulness for linking economic activity and industry energy demand, as the historical regional differences are subsumed by the regression of subsector energy demand on subsector activity, and the composition of subsector production is expected to move in the direction of higher shares of high-value products (decreasing physical production per unit value added) as economies evolve to higher GDP per capita, which acts in the same direction of increasing energy efficiency (decreasing energy demand per unit physical production), not introducing behaviour that is different from subsectors with physical representation.

The production of cement, chemicals, and iron and steel consumes the bulk of final energy in the industry sector (7 %, 14 %, and 23 %, respectively (Fischedick et al., 2014)), and accounts for 29 %, 13 %, and 30 % of total direct $CO_2$ emissions of industry (8.7 Gt $CO_2$/yr) (International Energy Agency, 2021). The application of CCS in industry has been studied for different subsectors (Kuramochi et al., 2012), but due to the limited number of key processes, the large average size of installations (which make for economic point sources) and the high specific $CO_2$ emissions by unit of economic output, the cement, chemicals, and steel subsectors are considered to be main targets for CCS in industry (Naims, 2016).

## 2.3 Electrification of Heat Production

Energy use in industry can be coarsely subdivided into two categories: energy for mechanical work, provided principally by electricity, and energy for heating, provided principally by fuels – especially on medium to high temperature levels (above 100°C). Both are represented in REMIND through individual nodes in the CES production structure of the industry subsectors, with low elasticities between them, as mechanical work and heat are not interchangeable in production processes. The substitution elasticities of fuels increases from low to high levels (0.5 or 0.7 to 2) by 2040, reflecting lower short-term and higher long-term flexibility in industry. Since the electrification of heat production is technically possible (Madeddu et al., 2020) and a viable option for mitigating $CO_2$ emissions (when using carbon-free electricity), we include an additional production factor for high-temperature heat from electricity in the production functions for both chemicals and other industry (cf. Figure 1).

This allows for the explicit modelling of the substitution between fuels and electricity in heat production, without allowing for the replacing of mechanical work through heat. This option is only included for the chemicals and other industry subsectors, since electric steel production is modelled explicitly (see section 3.3) and electrification of clinker burning (a major source of $CO_2$ emissions in cement production) is less attractive compared to CCS options, because it would only mitigate emissions

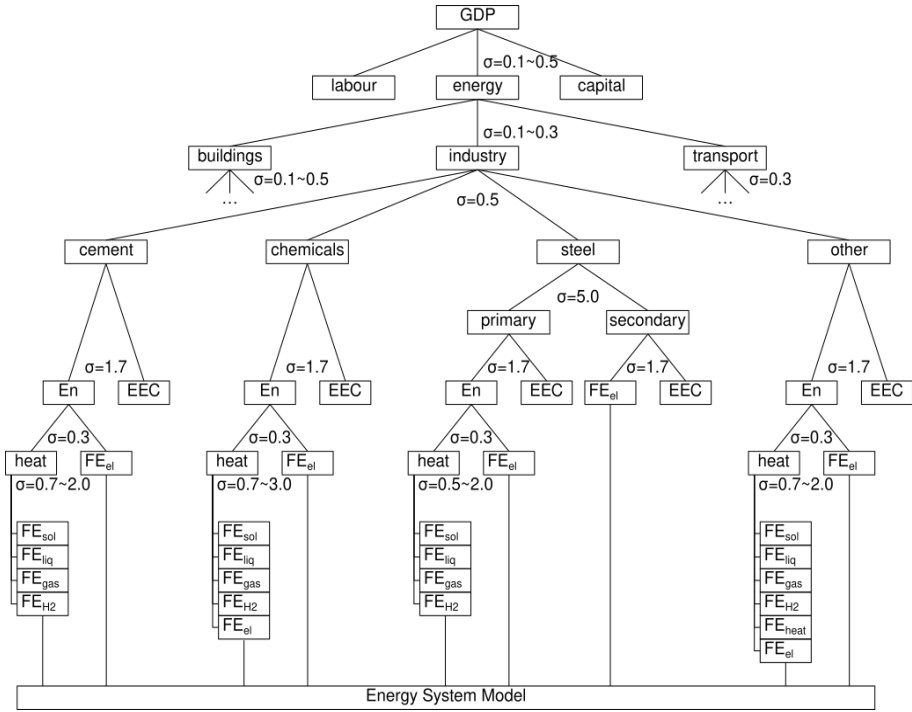

**Figure 1.** REMIND CES production tree. Macroeconomic output (GDP) is produced from labour, energy (energy services), and capital, while energy services are subdivided into the industry sector and the buildings and transport sectors (both ommited for clarity). Industry subsectors cement, chemicals, steel, and other industry, use final energy (FE) provided by the energy system model, and energy efficiency capital (EEC) to provide energy services. $\sigma$ denotes substitution elasticities between the respective inputs into a node, with $\sigma = a \sim b$ denoting elasticities changing over time.

from fuel burning, while large process emissions from limestone calcination (about half of total emissions) would remain (or require additional CCS).

The representation of energy carrier switching via the CES production function takes into account the heterogeneity of circumstances in industry subsectors, with some better positioned to electrify high-temperature heating than others.

## 2.4 Energy-Efficiency Investments

We also introduce dedicated stocks for energy efficiency capital (EEC) for all industry subsectors. They are positioned at the top level, such that subsector output is produced from the aggregated energy inputs and the EEC (cf. Figure 1). This models the trade-off between capital investment and energy demand. It is possible to invest into facilities with higher energy efficiency (usually for new installations, to some degree also for retrofits). This capital stock is integrated into the handling of the macroeconomic capital stock in REMIND, and is subject to depreciation and requires investments (Baumstark et al., 2021).

Both the stock of and the investments into capital for energy efficiency, separate from the general capital for industrial production, are difficult to ascertain (International Energy Agency, 2014). We therefore initialise this stock from investment estimates into energy-intensive (cement, chemicals, steel) and non-energy-intensive industry (other industry) for 2014–20 (International Energy Agency, 2014, Annex A), assuming a steady state where energy efficiency investments only cover the depreciation of the EEC stock, which is assumed to depreciate exponentially with a half-life of 25 years. For the baseline scenarios, we assume to EEC to grow (and shrink) proportional to subsector output (but EEC stocks are not reduced beyond the depreciation rate – in accordance with the capital motion formulation of the REMIND model).

## 2.5 Representation of Mitigation Options

Expanding on Kaya et al. (1997) and Fischedick et al. (2014), industry decarbonisation can be analysed using the identity

$$E = \text{Pop} \times \frac{\text{GDP}}{\text{Pop}} \times \frac{A}{\text{GDP}} \times \frac{\text{FE}}{A} \times \frac{\text{FF}}{\text{FE}} \times \frac{C}{\text{FF}} \times \frac{E}{C} \tag{2}$$

With $E$ the emissions, Pop the population, GDP the economic activity, $A$ the industrial production, FE the final energy use in industry, FF the fossil fuels used in industry, and $C$ the carbon content of those fossil fuels. The products of the identity are drivers of emissions or mitigation options: Pop – population growth (or population change more general), $\text{GDP}/\text{Pop}$ – per-capita GDP (affluence), $A/\text{GDP}$ – the share of industry in economic activity (as opposed to agriculture and services), $\text{FE}/A$ – the energy intensity of industry, $\text{FF}/\text{FE}$ – the share of fossil fuels in industry energy demand (as opposed to renewable energies), $C/\text{FF}$ – the specific carbon content of those fuels per unit energy (high for coal and oil, lower for natural gas), and $E/C$ – the emissions rate, which can be reduced by CCS.

Different drivers and mitigation options are realised by different parts of the extended REMIND model as presented here. Population is an exogenous SSP scenario assumption and thus constant across scenarios (that are based on the same SSP). GDP and industrial activity are endogenous elements of the production function and vary with the strictness of mitigation constraints. Final energy, fossil fuels, carbon content and emissions are all endogenous elements of the ESM. Notably, the REMIND model covers the entire range of mitigation options, from reduced demand for industrial goods, over increased energy efficiency in industry, fuel-switching and renewable power production, to CCS.

A forthcoming overview of industry $CO_2$ emission reduction (Bauer et al., in preparation) compares seven integrated assessment models "with improved industry sector representation", in which the REMIND model as discussed here shows the widest range of endogenous mitigation options. Notably, it is the only IAM with endogenous reduction of industry demand, due to its integration of a Ramsey growth model.

## 2.6 Implementation of Mitigation Options in REMIND

The role of the industry module is to provide the hard link between macroeconomic growth module and ESM, taking into account the aspects unique to industry (as opposed to the buildings and transport sectors). This entails (1) balancing industry final energy demand as an input to the CES production function with final energy provision by the ESM, (2) deriving industry

outputs and activity levels consistent with the overall macroeconomic developments and climate policy constraints, (3) deriving final energy demand consistent with industrial outputs and activity levels, (4) accounting for $CO_2$ emissions and options for their abatement via CCS, and consistency of industrial production with climate policy targets.

The first function is achieved by a simple balance equation that equates the sums of final energy inputs of different types (solids, liquids, gases, ...) into the different industry subsectors in the CES production function ($V_i$ in equation 1) with the production of those final energy carriers in the ESM. In that way, production of industry output that consumes energy incurs costs for energy production in the ESM, which have to be covered from the macroeconomic production (i.e. GDP output of the production function).

CES production functions, being an economic concept, do not represent all physical aspects relevant to modelling physical production in industry. They allow for the substitution of production factors beyond limits that might exist in real technical applications. Specifically, the formulation with industry subsector output being produced from energy and EEC captures the mechanism that through investments into energy efficiency, the specific energy demand per unit output can be reduced. But only up to a limit, given technical and physical (in the limit thermodynamic) constraints. We therefore impose a lower bound of the sum of final energy inputs into a subsectors' CES sub-tree per unit of subsector output, to confine the model solution space to technically and physically feasible values. This bound is described by an exponentially decreasing function that decreases from the 2015 specific energy demand towards the limit described below in section 3.6, and passes through a point that allows climate change mitigation scenarios to close no more than 75 % of the gap between the baseline scenario and this limit in 2050.

CES production functions also do not deviate far from equilibrium points. This is a problem in the detailed representation of final energy carriers with CES functions, as those carriers not already in use will see little utilisation even at very high price levels. This is most relevant for hydrogen and electricity for high-temperature heat production in industry, both of which are economically unattractive without strict climate policies and accompanying high carbon prices, and are therefore not used so far. To overcome this limitation, we employ two mechanisms.

First, we set future shares of hydrogen and high-temperature heat electricity for the baseline calibration. The share of hydrogen in industry gases is increased from 0.1 % in 2020 to 30 % in 2050. The share of high-temperature heat electricity in industry FE demand for high-temperature heat (so excluding electricity for mechanical work and low-temperature heat), is increased from 0.1 % in 2020 to 8 % in 2050. This leads to demand for hydrogen and high-temperature heat electricity in industry that is not in line with a "no climate change mitigation policy" baseline scenario, but necessary to generate realistic policy scenarios.

Second, we apply mark-up costs to both hydrogen and high-temperature heat electricity use in industry to represent additional cost of introducing new technologies to the production process that use these energy carriers.

Due to the economic nature of input substitution in the production function, the mark-up costs cannot be determined by techno-economic data and are instead set based on model behaviour. Both mechanisms, future baseline hydrogen/high-temperature heat electricity shares and mark-up costs, have been parametrised utilising the concept of the marginal rate of substitution, which describes the amount of one input needed to substitute another input to provide the same economic value (the ratio of the partial derivatives of two inputs into the production function). Final energy shares and mark-up cost are cho-

sen such that the marginal rates of substitution with respect to gases and liquids (as the final energy carriers hydrogen and high-temperature heat electricity compete most strongly with) roughly approach technical substitution ratios in climate policy scenarios (one for hydrogen, two to three in for high-temperature electricity). The mark-up costs can be reduced in scenarios which e.g. stipulate a strong policy push for these technologies (Schreyer et al., submitted), making hydrogen and/or high-temperature heat electricity cheaper relative to other final energy carriers and in turn increasing their utilisation in industry.

Finally, we impose a lower bound on the share of steel from primary production (cf. section 3.3 below).

Industry CCS is calculated applying subsector-specific marginal abatement cost (MAC) curves. The MAC curves are either based on (Kuramochi et al., 2012), a techno-economic assessment of $CO_2$ capture technologies (with capture rates of 28–76 % at costs of 62–133 $\$US_{2005}$/t $CO_2$[3] for different subsectors), or on (Fischedick et al., 2014, fig. 10.7–10.10), implementing a more optimistic industry CCS scenario (with capture rates of 75 % at 50 $\$US_{2005}$/t $CO_2$ up to 95 % at 217 $\$US_{2005}$/t $CO_2$ for all subsectors except "other industry"). Subsector emissions from fuel combustion are calculated from final energy demand and fuel-specific emission factors. Process emissions in the cement subsector are calculated by applying a specific emissions factor for clinker (0.53 t $CO_2$/t clinker) and regional clinker to cement ratios (0.58-0.82 t clinker/t cement; Kermeli et al. (2016), fig. 21), which are converged towards to lowest regional value by 2100. The model is then able to capture industry $CO_2$ emissions up to the level indicated by the respective MAC curve at given $CO_2$ prices. Since the MAC curves do not reproduce the inertia of capital stocks required for CCS (investments and retrofits would occur only gradually, not all at once), the increase in the capture rate is limited to 5 % p.a.

## 3  Projections and Input Data

### 3.1  General Approach

Since there is structural uncertainty in the future development of the fundamental drivers of the energy system – population and GDP growth – IAMs use the Shared Socioeconomic Pathways (SSPs) as a common framework (O'Neill et al., 2017). The five SSPs (named SSP1 through SSP5) provide both a contextual description of how the world might evolve in the coming century, and quantitative trajectories for future population and GDP on the country level. They are varied along two orthogonal axes: socioeconomic challenges to climate change mitigation, and socioeconomic challenges to adaptation to climate change. In this paper, we consider the scenarios SSP1, SSP2, and SSP5, updates from earlier SSP-realisations with the REMIND model (Kriegler et al., 2017).

- The SSP2 scenario is characterised by median challenges along both axes and referred to as a "middle of the road" scenario, where historic trends continue into the future, improving living conditions for most people without pronounced reductions in global inequality, and technological progress at a steady but constant rate.
- The SSP1 scenario is characterised by low challenges along both the mitigation and adaptation axes and referred to as a "sustainability" scenario, where the demographic transition is accelerated, leading to lower population growth, and

---

[3]All monetary values and prices in REMIND are denoted in year 2005 US-dollars and converted accordingly.

high-income countries shift to "a broader emphasis on human well-being", leading to slower economic growth, and investments and adjusted tax incentives bring about faster progress in resource efficiency.

– The SSP5 scenario is characterised by high challenges to mitigation, but low challenges to adaptation, and referred to as a "fossil fuel-development" scenario, where the exploitation of abundant fossil fuels drives investment into health and education, but also consumption, and therefore very energy-intensive lifestyles globally.

We project industry subsector activity, as well as final energy demand, for the SSP2 ("middle of the road") scenario, which best fits the continuation of historic trends into the future. All other scenarios, including SSP1 and SSP5, are derived as variations of specific material and specific energy demand of the SSP2 scenario.

Industry subsector activity (except for steel production) for the SSP2 scenario is projected in per-capita terms as a function of per-capita GDP in the form of

$$A_{\mathrm{pC}} = \alpha \exp\left(-\frac{\beta}{\mathrm{GDP}_{\mathrm{pC}}}\right) \tag{3}$$

with $A_{\mathrm{pC}}$ the per-capita activity level, $\mathrm{GDP}_{\mathrm{pC}}$ the per-capita GDP, $\alpha$ the asymptotic limit of $A_{\mathrm{pC}}$, and $\beta$ a parameter of convergence speed. This formulation is used because it presupposes a decoupling of per-capita demand from increasing 275 affluence levels, and its positive codomain makes it easily tractable. It was found to be a good fit to historic data in previous research (van Ruijven et al., 2016). A decoupling between GDP and production of bulk industrial goods (cement and steel) or the value added of industry (chemicals and other industry) is in line with historically observed patterns, and generally assumed to continue as future economic activity moves from physical production to service provision (Bashmakov et al., 2022). Parameters $\alpha$ and $\beta$ are derived via regression both on regional and global levels. The regional asymptotes $\alpha$ are 280 linearly converged towards the global value until 2200 (i.e. in about 2100 they lie half-way between the original regional and the global levels). This results in a short-term continuation of current trends and a long-term convergence across regions. Using these converged parameters, per-capita and absolute activity are projected using per-capita GDP and population projections (KC and Lutz, 2017, Koch and Leimbach (2023)).

For cement, activity levels are derived in physical units (production quantity), for the chemical industry they are expressed 285 in economic units (subsector value added), due to the heterogeneity of chemicals produced. Since cement and steel production are modelled in physical units, an additional regression of value added per unit production is performed (see subsections 3.5. This allows calculating consistent estimates for the "other industry" subsector as the difference between total industry value added and the value added of the cement, chemicals, and steel subsectors. These regressions use the same formula as used for per-capita output (equation 3), presupposing an increasing specific value added (dollars per unit of production) that converges 290 toward some level, as industry moves to production of increasingly higher-value products (e.g. high-strength or corrosion resistant steels rather than generic construction steels). As before, the regression is performed both on the regional and global level, and the regional asymptotic limits $\alpha$ are linearly converged towards the global value until 2200 (i.e. in about 2100 lie half-way between the original regional and global levels). Population and GDP regression data is from World Bank (2019),

**Table 1.** Data sources for input data regression and projection

| Subsector | Variable | Source |
|---|---|---|
| all | historic population | United Nations Department of Economic and Social Affairs Population Division (2015) |
| | historic per-capita GDP | World Bank (2019), series NY.GDP.MKTP.PP.KD ("GDP, PPP; constant 2017 international $") and SP.POP.TOTL ("Population, total"), from 1970 through 2021 |
| | | Arujo et al. (2021) |
| | population projections | KC and Lutz (2017) |
| | | Koch and Leimbach (2023) |
| | per-capita GDP projections | James et al. (2012) |
| cement | cement production | van Ruijven et al. (2016) |
| | | van Ruijven (2017) |
| | | USGS (2020) |
| steel | in-use steel stocks | Müller et al. (2013), supplementary information 2, sheet "steel stock per cap med" |
| | steel stock life times | Pauliuk et al. (2013b) (table S23, approach b) |
| chemicals | chemicals value added | United Nations Industrial Development Organization (2017), ISIC Rev. 3 code 24 (chemicals and chemical products) |
| other industry | industry value added | United Nations Industrial Development Organization (2017), using ISIC Rev. 3 code D "total manufacturing" and table code 20 "value added" |

series NY.GDP.MKTP.PP.KD ("GDP, PPP; constant 2017 international $") and SP.POP.TOTL ("Population, total"), from 1970 through 2021. Table 1 collects all data sources used for all or for individual subsector regressions and projections.

### 3.2 Cement Projections

Historic per-capita cement production is regressed on per-capita GDP using the formulation discussed above (equation 3), using cement production summaries by van Ruijven et al. (2016), van Ruijven (2017), and USGS (2020). Regression parameters are derived once on the regional level, and once again on the global level. For the global regression, cement production from the "CHA" region (Peoples' Republic of China with Hong Kong and Macau, and Republic of China) is excluded, since the exceptional cement production of the People's Republic (55 % of global production at 18 % of global population in 2019) would unduly bias the global regression. Projections of per-capita cement production and total cement production are calculated from the converged regression parameters (see above) and per-capita GDP and population projections for the SSP2 scenario, respectively. Figure 2 shows the regression data (regionally aggregated), the regression function, and the SSP2 projection.

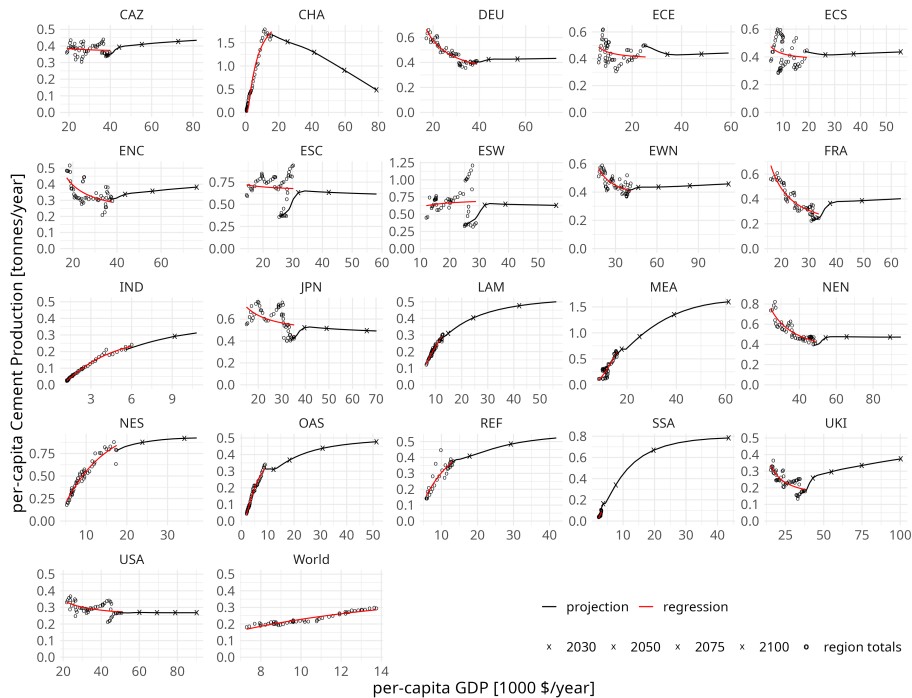

**Figure 2.** Regression and projection of cement production. Per-capita cement production over per-capita GDP for 21 REMIND regions (see Appendix A).

## 3.3 Steel Projections

Iron and Steel differ from other bulk materials like cement and basic chemicals by being easily recyclable. Cement is bound in concrete with aggregate and possible reinforcement that requires elaborate separation and energy intensive reversal of the hydration reactions during setting to be recycled. Chemicals like ethylene and propylene are processed into complex products like plastics that are widely dispersed in the economy, requiring careful separation from other materials and energy intensive treatments to recover feedstocks or products that are often not of the comparable quality as the ones being recycled (Uekert et al., 2023). Steel can mostly be collected, separated magnetically from other materials, and molten down to produce new products. (There are important constraints to the quality of recycled steel, due to tramp elements like copper (Daehn et al., 2017), currently limiting the share of recycled steel.) For these reasons, steel, unlike cement and chemicals, accumulates in the economy. Most iron ore that has been refined into steel stays available as steel.

Our projections of steel production therefore follow the approach of Pauliuk et al. (2013a), which considers the (per-capita) stock of steel in use in the economy and not the yearly flow of new steel being produced as primary regression variable. It assumes that in-use per-capita steel stocks will saturate over time and that deprecated stocks will then be replaced mainly by recycled steel. Projections of steel production derive from carrying forward the steel stocks, calculating yearly stock losses from deterioration to scrap, and based on available scrap calculating secondary and then primary steel production.

Estimates of in-use steel stocks per country (Müller et al., 2013, supplementary information 2, sheet "steel stock per cap med") are aggregated to 21 regions and converted to per-capita numbers with population data (United Nations Department of Economic and Social Affairs Population Division, 2015) and then regressed on per-capita GDP (World Bank, 2019, Arujo et al. (2021)) using the logistic function

$$S_{\mathrm{pC}} = \frac{A}{1 + \exp\left(\frac{i - \mathrm{GDP}_{\mathrm{pC}}}{s}\right)} \tag{4}$$

with $S_{\mathrm{pC}}$ the per-capita steel stock, $\mathrm{GDP}_{\mathrm{pC}}$ per-capita GDP, $A$ the saturation level (asymptote) of per-capita steel stock, $i$ the per-capita GDP level at which the steel stock is half as much as the asymptote (inflection point), and $s$ a scaling parameter determining the speed of convergence. The resulting regression parameters $A$, $i$, and $s$ are used with per-capita GDP projections from James et al. (2012) and population projections from KC and Lutz (2017) as well as Koch and Leimbach (2023) to project future per-capita and absolute steel stocks.

Projected steel stocks are depreciated exponentially with steel stock life-times $l$ from Pauliuk et al. (2013b) (table S23, approach b), which are converged linearly towards the global average in 2100. Of the depreciated steel stocks, we assume 90% to be available for recycling. The resulting year-on-year differences in steel stocks (due to both per-capita steel stocks increasing monotonically with increasing per-capita GDP, and replacements for depreciated stocks) are the required stock additions. Decreases in steel stocks due to shrinking populations are not balanced, but ignored, as they would otherwise imply a reduced

per-capita steel production in regions with contracting populations. (The life-times of steel stocks are largely independent of utilisation and thus population. Buildings, transport equipment and goods do not become obsolete at a lower rate because there are more of them per capita, they are written off just the same.)

The stock additions from increasing per-capita stocks and increasing population, and the replacements for depreciated stocks are added up to derive steel demand, and, adjusted for trade, domestic steel production in each region. Steel and steel products

are widely traded commodities, but modelling or projecting future steel trade is beyond the scope of our model. It is therefore assumed that trade patterns in the base year continue over the model horizon. To that end, regional net trade ($T_r$) (positive for net imports, negative for net exports) is kept at the same share in regional steel use (production plus net trade) over time as in the base year. Since the sum of these trade volumes across regions is not generally equal to zero after the base year, they are adjusted ($T_r'$) by scaling all imports and exports inversely to their share in total trade (global net trade – positive if all imports

exceed all exports and vice versa – over the global sum of absolute trade flows). Thus if the global sum of imports is twice as large as that of exports, the imbalance is solved by scaling imports down by a factor twice as large as the factor with which exports are scaled up.

$$T_r' = T_r \left( 1 - \mathrm{sign}\left(T_r\right) \frac{\sum_{r'} T_{r'}}{\sum_{r'} |T_{r'}|} \right) \tag{5}$$

Steel demand adjusted for trade is served by secondary production first, subject to a scrap availability constraint (share of

recyclable steel from the deprecated steel stock) and a constraint on the maximum (minimum) share of secondary (primary)

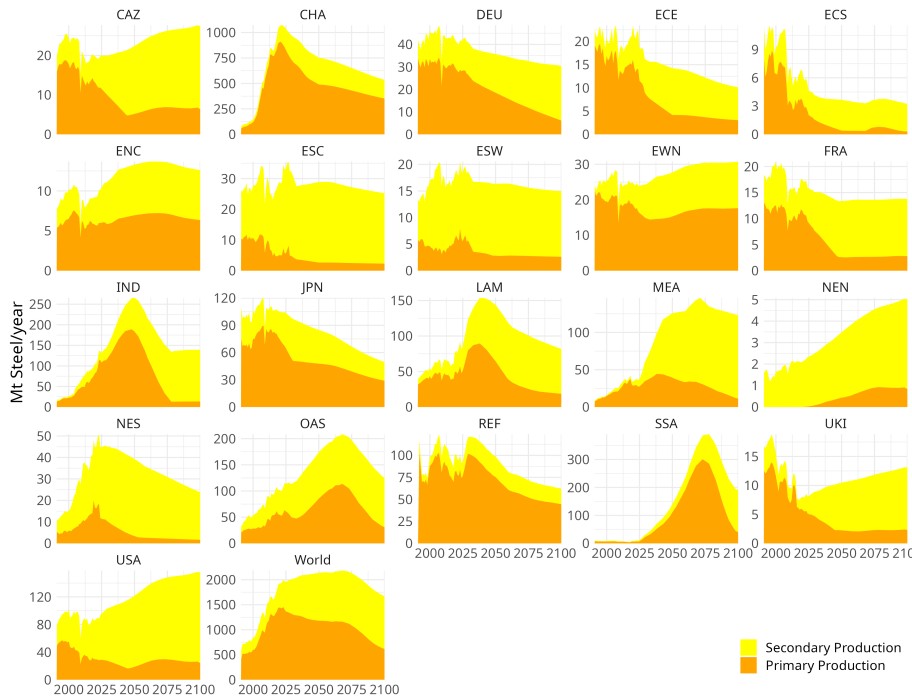

**Figure 3.** Primary and secondary steel production. Historic production unit 2019 and projections for the SSP2 scenario for 21 REMIND regions (see Appendix A).

steel production, as we assume secondary production to be cheaper than primary production. The remainder of total production is then primary production. The maximum share of secondary steel production is capped at a value that converges linearly from base year levels to the assumed bound of 10 % in 2050, if no region-specific information is used. The reason is twofold: (1) certain applications require high quality steel that can only be achieved via primary production, (2) we assume that current production practices and existing production capacity will not change quickly, but continue to be used for economic reasons. Figure 3 shows the historic production (until 2019) and SSP2 projections of primary and secondary steel production.

### 3.4 Chemical Projections

Historic per-capita value added of chemicals production is regressed on per-capita GDP using equation 3. Chemicals value added data is from United Nations Industrial Development Organization (2017), ISIC Rev. 3 code 24 (chemicals and chemical products), per-capita GDP data from World Bank (2019) and population data from United Nations Department of Economic and Social Affairs Population Division (2015). Regression parameters are derived once on the regional level, and once again on the global level, and regional asymptotes are converged linearly towards the global ones by 2200. Projections of per-capita chemical value added and total chemicals value added are calculated from the converged regression parameters (see above)

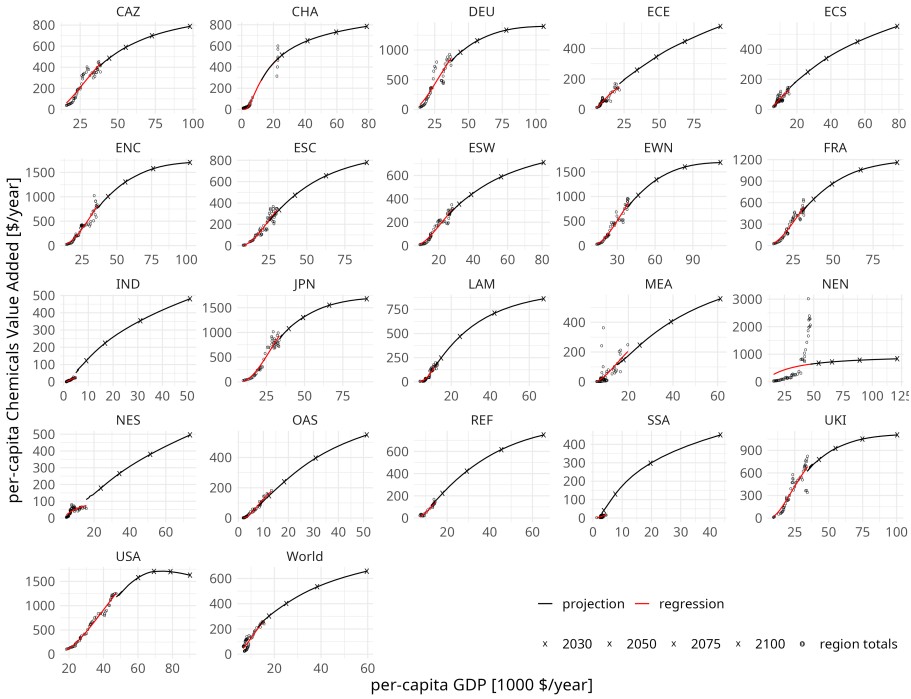

**Figure 4.** Regression and projection of chemicals value added. Per-capita chemicals value added over per-capita GDP for 21 REMIND regions (see Appendix A).

and per-capita GDP and population projections for the SSP2 scenario, respectively. Figure 4 shows the chemicals regression
data (regionally aggregated), the regression function, and the SSP2 projection of chemicals value added.

### 3.5  Other Industry Projections

Projecting the activity (value added) of the "other industry" subsector entails projecting the value added of industry as a whole
and subtracting the value added of the three subsectors that are modelled explicitly. To that end, per-capita value added of
the entire industry sector is regressed on per-capita GDP using equation 3. Industry value added data is from United Nations
Industrial Development Organization (2017), using ISIC Rev. 3 code D "total manufacturing" and table code 20 "value added",
per-capita GDP from World Bank (2019) and population data from United Nations Department of Economic and Social Af-
fairs Population Division (2015). Regression parameters are derived once on the regional level, and once again on the global
level, and regional asymptotes are converged linearly towards the global ones by 2200. Figure  5 shows industry value added
regression data (regionally aggregated), the regression function, and the value added projections.
Specific value added of cement and steel production (dollars value added per tonne of physical production) are regressed
on per-capita GDP using equation  3. Cement and steel value added data is from United Nations Industrial Development
Organization (2017), ISIC Rev. 3 codes 26 "non-metallic mineral products" (cement) and 27 "basic metals" (steel), respectively,

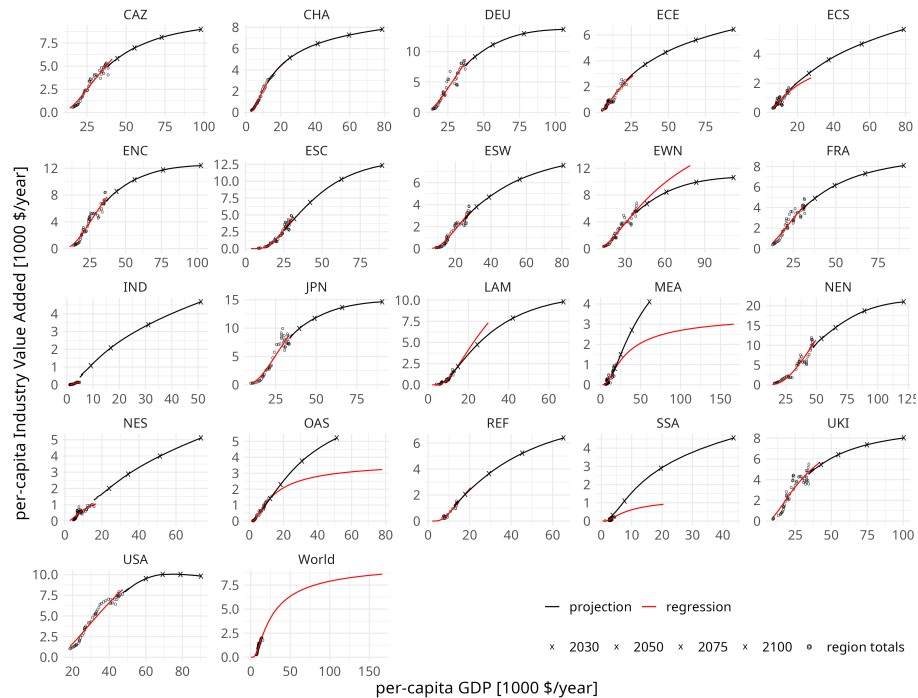

**Figure 5.** Regression and projection of total industry value added. Per-capita industry value added over per-capita GDP for 21 REMIND regions (see Appendix A).

and table code 20 "value added". Cement production data is from van Ruijven et al. (2016) and van Ruijven (2017), steel production data from World Steel Association (2018). Regression parameters are derived on the regional and global level, and

regional asymptotes are converged linearly towards the global one by 2200. Using the converged regression parameters for cement and steel specific value added and the projected production of cement and steel, (total) value added of cement and steel production is projected. Figures 6 and 7 show the specific cement and steel regression data (regionally aggregated), the regression functions, and the value added projections.

Lastly, projected value added of cement, chemicals, and steel production is subtracted from projected value added of the

entire industry sector to yield projected value added of the "other industry" subsector. Figure 8 shows the absolute value added of all four industry subsectors for the SSP2 scenario.

## 3.6 Final Energy Projections

Demand of final energy is projected using annual autonomous energy efficiency improvements (Kermeli et al., 2014) that decrease the total (i.e. across energy carriers) specific final energy demand (Joule per unit of production or per unit value added)

towards a lower bound, and by converging the shares of different final energy carriers in total specific final energy demand from

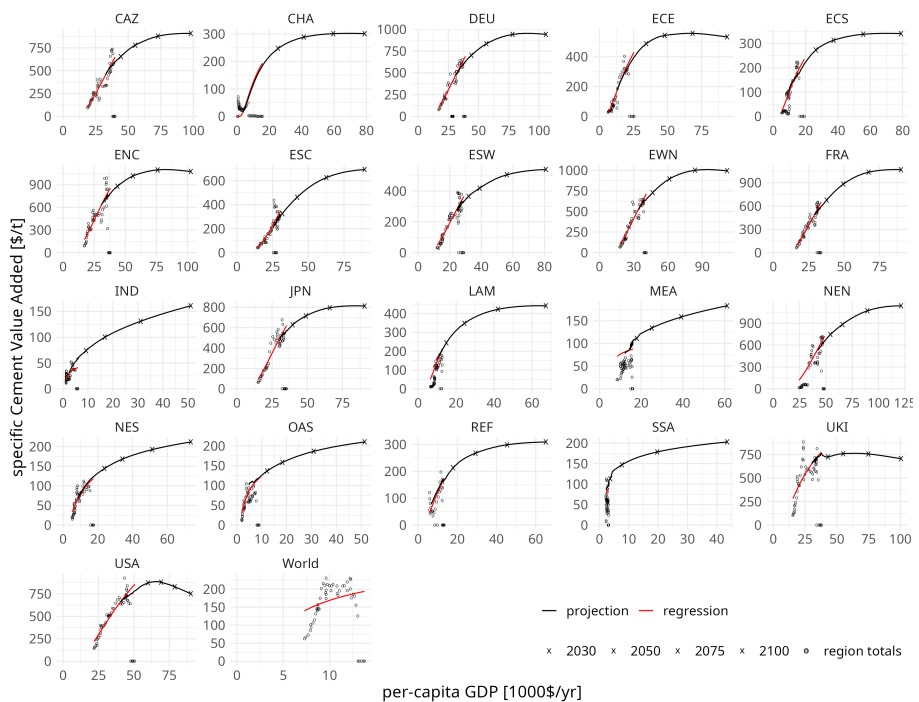

**Figure 6.** Regression and projection of specific cement value added. Per-tonne cement value added over per-capita GDP for 21 REMIND regions (see Appendix A).

historic values towards those in International Energy Agency (2017) in 2060 (and keeping them constant afterwards). Total (subsector) specific final energy demand is calculated according to the formula

$$\frac{E(t)}{A(t)} = \left(\frac{E(t_0)}{A(t_0)} - L\right)(1-\alpha)^{t-t_0} + L \tag{6}$$

with $E$ the final energy demand, $A$ the activity level (physical production or value added), $L$ the lower limit of specific
final energy demand, $\alpha$ the annual autonomous energy efficiency improvements, and $t_0$ the base year. The annual autonomous energy efficiency improvements are chosen such that overall industry final energy demand trajectories conform to scenario assumptions by other groups (International Energy Agency, 2017). The limits $L$ are set at 1.8 GJ/t for cement (Madlool et al., 2011), 8 GJ/t and 1.3 GJ/t for primary and secondary steel (Fruehan et al., 2000), and 10 % of the base-year value of the region with the lowest specific energy demand (final energy per value added) for chemicals and other industry. Table 2 summarises
the parameters for the specific final energy demand projections.

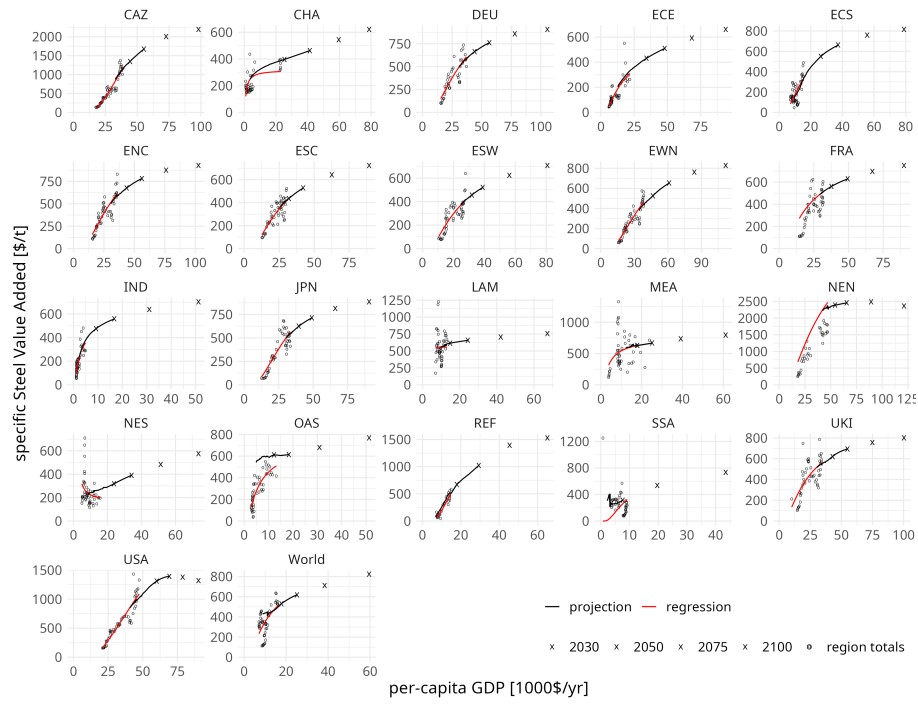

**Figure 7.** Regression and projection of specific steel value added. Per-tonne steel value added over per-capita GDP for 21 REMIND regions (see Appendix A).

**Table 2.** Convergence parameters for specfic FE demand

| Subsector | Lower Limit | Lower Limit Source | $\alpha$ |
|---|---|---|---|
| cement | 1.8 GJ/t | Madlool et al. (2011) | 2.1 % |
| primary steel | 8 GJ/t | Fruehan et al. (2000) | 0.48 % |
| secondary steel | 1.3 GJ/t | Fruehan et al. (2000) | 0.48 % |
| chemicals | 10 % of base year | assumption | 1.6 % |
| other industry | 10 % of base year | assumption | 2.1 % |

## 3.7 Scenario Variations

Other scenarios are derived from the SSP2 scenario by varying specific material demand (subsector production $A$ per unit GDP) and specific energy demand (Joule per unit subsector production). Three methods for varying specific material demand are used: "fixed ratios", "declining improvements", and "modified rate of change", which calculate the scenarios' specific material demand by multiplying either the specific material demand of a reference scenario $R$ ("fixed ratios" and "declining


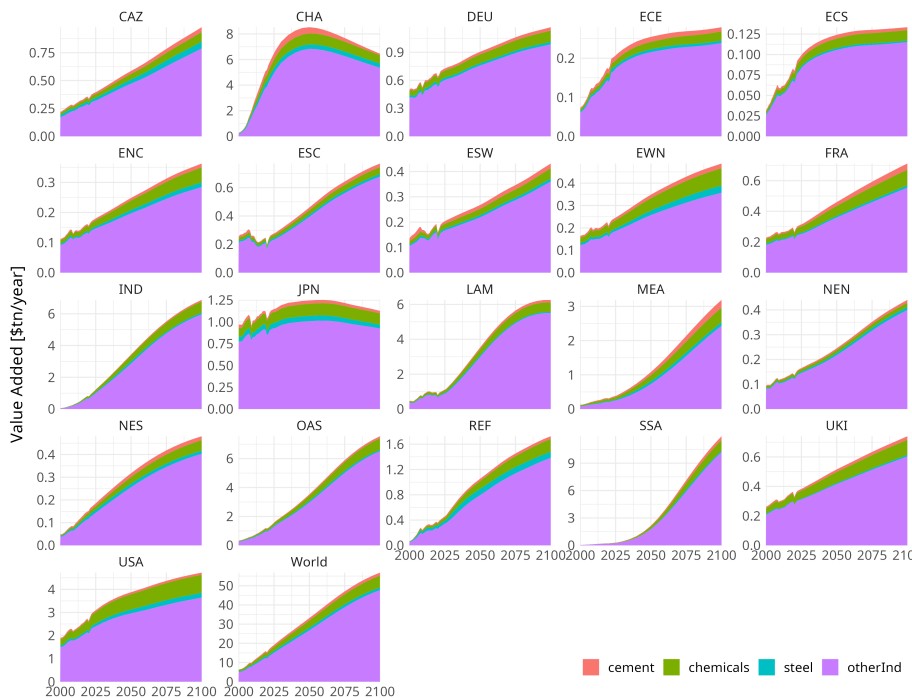

**Figure 8.** Industry subsector value added. Historical data and projections of value added of the cement, chemicals, steel, and other industry subsectors for the SSP2 scenario for 21 REMIND regions (see Appendix A).

improvements"), or the specific material demand of base year $t_0$ ("modified rate of change", base year specific material demand is identical across scenarios) with a factor $\alpha(t)$, which is derived differently for the three methods.

Under the "fixed ratios" method, specific material demand is set to a specific ratio $\alpha_0$ of the specific material demand of the reference scenario $R$ (e.g. cement demand per unit GDP can be set to 95 % of specific cement demand in the reference scenario). The actual parameters $\alpha(t)$ is converged from one to $\alpha_0$ over a time period $T_C$ (15 years) from the base year $t_0$ on.

$$\frac{A_s(t)}{\text{GDP}_s(t)} = \alpha(t)\frac{A_R(t)}{\text{GDP}_R(t)} \tag{7}$$

$$\alpha(t) = 1 + (\alpha_0 - 1)\min\left(1, \max\left(0, \frac{t - t_0}{T_C}\right)\right) \tag{8}$$

Under the "declining improvements" method, annual improvements of specific material demand are assumed, that converge from $\alpha_0$ (e.g. 0.5 % per annum decrease of cement demand per unit GDP) to zero over the configurable time period $T_C$ from base year $t_0$ on (to preclude exponential decay of the specific demand towards zero). The parameter $\alpha(t)$ is the product of converged $\alpha_0$ parameters from the base year to year $t$, or one before the base year.

$$\frac{A_s(t)}{\text{GDP}_s(t)} = \alpha(t) \frac{A_R(t)}{\text{GDP}_R(t)} \tag{9}$$

$$\alpha(t) = \begin{cases} 1 & t \le t_0 \\ \prod_{r=t}^{t+T_C} 1 - \alpha_0 \left(1 - \min\left(1, \frac{t-t_0}{T_C}\right)\right) & t > t_0 \end{cases} \tag{10}$$

Under the "modified rates of change" method, relative differences in specific material demand (with respect to the base year $t_0$) of the reference scenario $R$ are either improved (reductions are turned into higher reductions, increases are turned into lower increases) or degraded (increases are turned into higher increases, reductions are turned into lower reductions). To that end, the relative difference $\beta(t)$ of specific material demand of the reference scenario $R$, relative to the base year $t_0$

$$\beta(t) = \frac{A_R(t)}{\text{GDP}_R(t)} \frac{\text{GDP}_R(t_0)}{A_R(t_0)} - 1 \tag{11}$$

is multiplied by some factor $\alpha_0$ if the difference is positive (increasing specific demand), or divided by $\alpha_0$ if the difference is negative (decreasing specific demand) to calculate modified relative differences $\alpha(t)$

$$\alpha(t) = \begin{cases} 0 & t \le t_0 \\ \beta(t)\alpha_0^{-\text{sign}(\beta(t))} & t > t_0 \end{cases} \tag{12}$$

and the specific material demand of scenario $S$ is calculated using base year specific material demand and these modified relative differences

$$\frac{A_S(t)}{\text{GDP}_S(t)} = (1 + \alpha(t)) \frac{A_S(t_0)}{\text{GDP}_S(t_0)} \tag{13}$$

The method of calculating specific material demand, as well as the $\alpha_0$ parameters (and $T_C$ in the case of "declining improvements") can be chosen individually for combinations of scenarios, regions, and industry subsectors. Table 3 summarises the settings used to derive the SSP1 and SSP5 scenarios from the SSP2 scenario.

**Table 3.** Parameters for deriving the SSP1 and SSP5 scenarios from the SSP2 scenario

| Scenario | Subsector | Region | Method | $\alpha_0$ | $T_C$ |
|---|---|---|---|---|---|
| SSP1 | cement | CHA | declining improvements | 6.5 % p.a. | 50 years |
| SSP1 | cement | others | declining improvements | 3 % p.a. | 50 years |
| SSP1 | chemicals | all | declining improvements | 5 % p.a. | 50 years |
| SSP1 | primary steel | CHA | declining improvements | 10 % p.a. | 50 years |
| SSP1 | primary steel | others | declining improvements | 6 % p.a. | 50 years |
| SSP1 | secondary steel | CHA | declining improvements | 10 % p.a. | 50 years |
| SSP1 | secondary steel | others | declining improvements | 6 % p.a. | 50 years |
| SSP1 | other industry | all | declining improvements | 2 % p.a. | 50 years |
| SSP5 | all | all | modified rates of change | 0.5 | — |

## 4  Industry Demand in the System Context: Exemplary results

This section exemplifies industry subsector results from the REMIND model for different scenarios. We examine the three socioeconomic baselines SSP1, SSP2, and SSP5, and combine them with three different policy scenarios. The baseline scenarios (Base) reproduce the basic SSP scenario narratives without any climate change mitigation policies, and are identical to the calibration data derived as described in section 3. The two mitigation scenarios use carbon price trajectories that result in cumulated GHG emissions (i.e. the carbon budget) from 2020 on to peak at 1150 and 500 Gt $CO_2$-equivalent ("PkBudg1150" and "PkBudg500") and subsequently decline through net-negative emissions, which are consistent with two possible interpretations of the climate targets of the Paris Accord (limiting global warming to 2°C with 67 % likelihood, or to 1.5°C with 50 % likelihood, cf. Arias et al. (2021), table TS.3).

Figure 9 shows industry subsector activity, indexed to the year 2015. We observe differentiated demand reactions to climate policy across the socioeconomic baselines, the policy scenarios, as well as between different industry subsectors.

In accordance with the SSP scenario narratives (O'Neill et al., 2017, Bauer et al. (2017)) (see section 3.7), the SSP1 Base scenario shows lower growth in industry demand (-30 to 210 %, 2015 to 2100 change) than the SSP2 scenario (-47 to 360 %), while the SSP5 Base scenario shows higher growth (140 to 810 %), reproducing the data from the projection and scenario variation steps (section 3). The "other industry" subsector sees the highest growth across all socioeconomic baselines (210 to 810 %), in line with the assumed increase in the share of industries producing high-value goods as opposed to bulk materials. The increase in the chemicals subsector's activity (46 to 690 %), too, agrees with the assumption of higher shares of products from light metals like aluminium, plastics and other modern materials, rather than steel.

For all socioeconomic baselines, we see a reduction in industry demand with strengthened climate targets (reduced budgets for further GHG emissions), as well as a shift from primary (-47 to 140 %) to secondary steel (4 to 610 %), since the latter is produced from electricity which is easier to produce carbon-free.

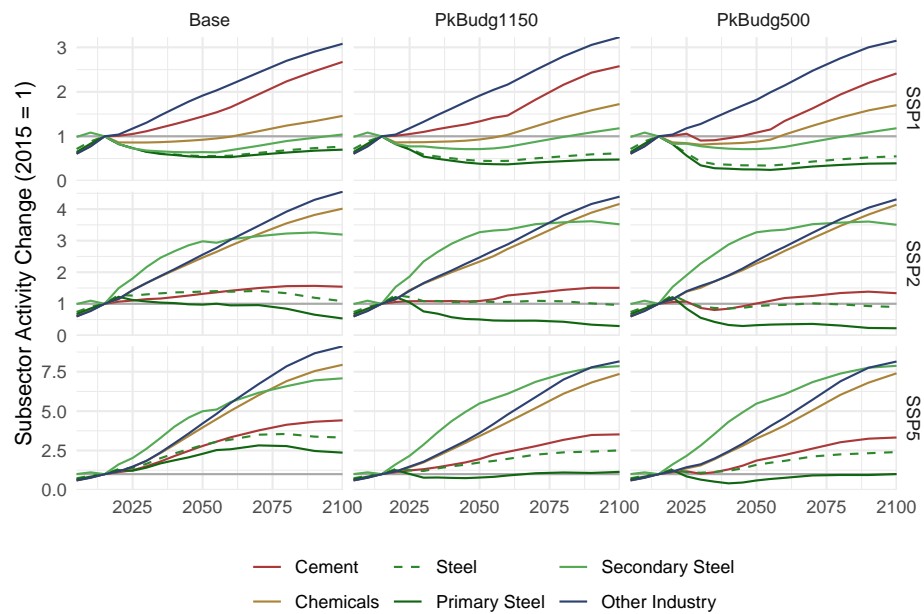

**Figure 9.** Industry Subsector Activity Change. Production of cement and primary and secondary steel, as well as chemicals and other industry value added, indexed to 2015. Panels correspond to scenario matrix, with socioeconomic drivers (SSP1, SSP2, SSP5) and climate change mitigation policy (Base – no policy, PkBudg1150 and PkBudg500 – peak budgets of 1150 and 500 GtCO$_2$, corresponding to the 2°C and 1.5°C targets.)

Figures 10 and 11 show industry final energy (FE) demand, by subsector and energy carrier. Both the shares of individual
energy carriers, as well as of the different subsectors, vary across the scenario matrix, as the model reacts differently to the emission constraints in the different socioeconomic baselines.

For a comparison of total industry FE demand with historic data and other projections, see Appendix B. Total FE demand of the SSP2 scenario increases in accordance with the "Reference Technology Scenario" from International Energy Agency (2017) until 2050 (cf. section 3.6), with SSP1 and SSP5 markedly higher and lower, respectively.

The SSP1 Base scenario shows an immediate decline in industry FE demand, in accordance with the scenario narrative of sustainability and efficiency, while the SSP2 and SSP5 scenarios display an increase and subsequent decrease in FE demand, albeit to markedly different levels (273 EJ/year peak in 2070 for SSP2, and 559 EJ/year peak in 2080 for SSP5). The share of electricity (an indicator of industry decarbonisation) rises to 28 % in 2090 for the Base scenarios, and 45–49 % in 2080 (PkBudg1150) and 2055 (PkBudg500) for the mitigation scenarios, while the share of solids decreases strongly, to be partially
replaced with liquid final energy carriers, part of which will be generated from biomass.

Figure 12 exemplifies the differences in mitigation strategies of the industry subsectors for the set of SSP2 scenarios. The cement and chemicals subsectors, which cannot easily electrify, move to carbon-free fuels either directly in the from of biomass or hydrogen, or indirectly by using hydrogen-derived synthetic fuels (biomass and synthetic fuel are grouped under "carbon free"). The steel sector decarbonises by expanding production of secondary steel that uses carbon-free electricity (within the

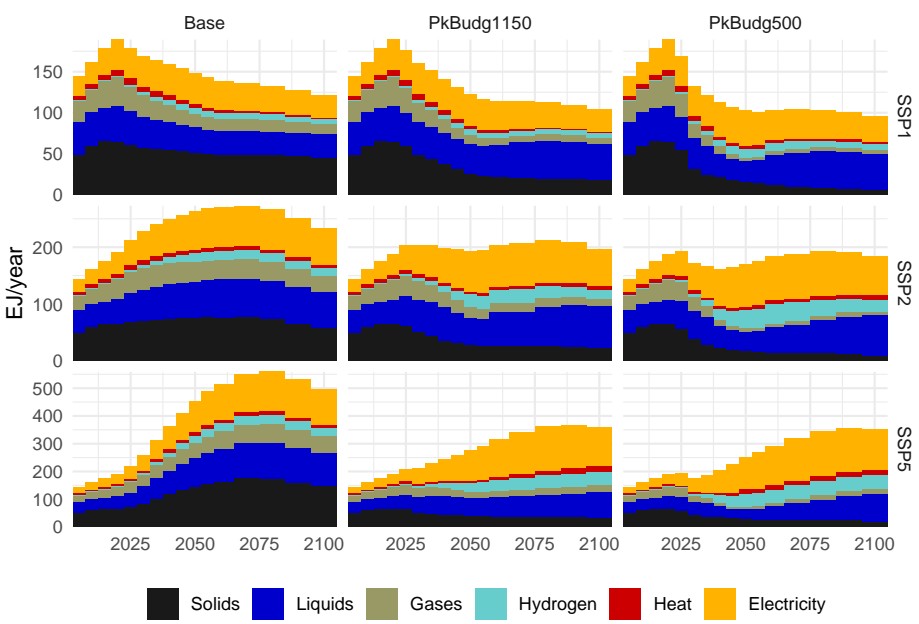

**Figure 10.** Industry final energy demand by energy carrier. Panels correspond to scenario matrix, with socioeconomic drivers (SSP1, SSP2, SSP5) and climate change mitigation policy (Base – no policy, PkBudg1150 and PkBudg500 – peak budgets of 1150 and 500 $GtCO_2$, corresponding to the 2°C and 1.5°C targets.). Solids, Liquids, and Gases combine final energy carriers from fossil and non-fossil sources.

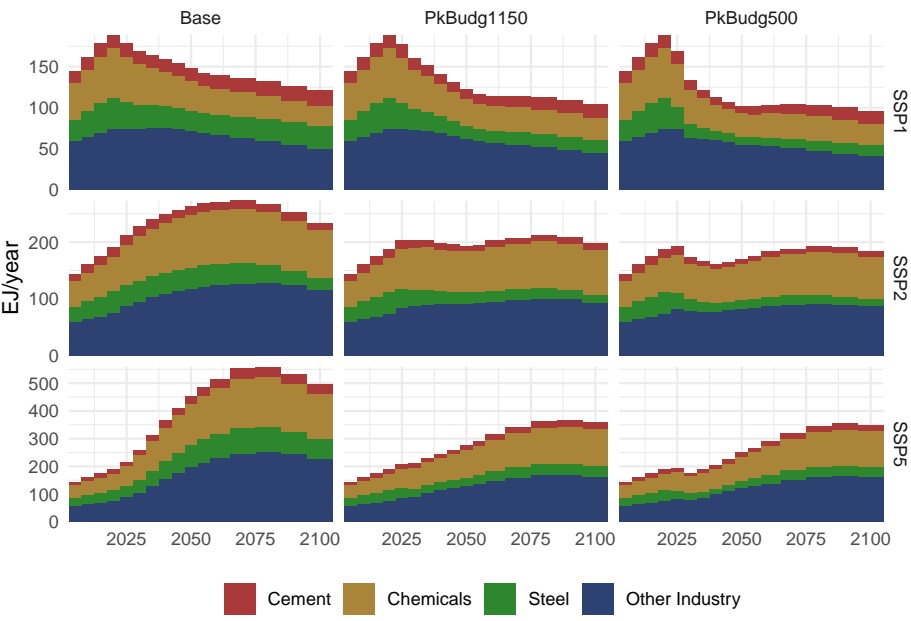

**Figure 11.** Industry subsector final energy demand. Panels correspond to scenario matrix, with socioeconomic drivers (SSP1, SSP2, SSP5) and climate change mitigation policy (Base – no policy, PkBudg1150 and PkBudg500 – peak budgets of 1150 and 500 $GtCO_2$, corresponding to the 2°C and 1.5°C targets.)

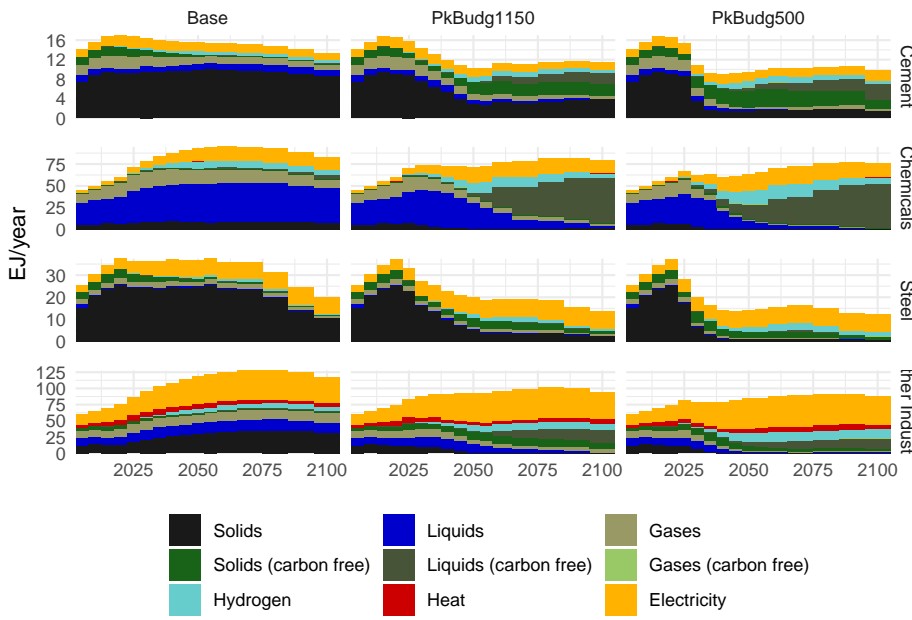

**Figure 12.** Industry final energy demand by subsector and energy carrier for SSP2 scenario. Solids, liquids, and gases derived from either biomass or (renewable) hydrogen are grouped as "carbon free".

limits of scrap availability), by reducing production of primary steel, and by switching to carbon-free fuels. The other industry subsector substitutes electricity for fuels, while also increasing energy efficiency, so the total electricity demand increases only slightly, and moves to carbon-free fuels. Notably, the residual share of fossil fuels is lowest in the other industry subsector, as it does not allow for the application of CCS.

Figure 13 shows industry $CO_2$ emissions and CCS by industry subsector. The emissions in the baseline scenarios follow the trajectory of the total demand in final energy of fossil origin (cf. solids, liquids, and gases in Figure 10), while their composition differs from the subsector-composition of FE demand (cf. Figure 11, as the other industry subsector moves to higher shares of electricity in FE demand, while the heavy industry subsectors rely more strongly on fossil fuels for high-temperature heat generation. Process emissions from cement production become a dominant fraction, as they are fixed relative to cement production, thus do not benefit for energy efficiency improvements, and cement production expands (especially in emerging economies). The electrification trend is intensified in the climate mitigation scenarios and complemented by CCS to mitigate emissions from fossil fuels in the heavy industry subsectors that cannot be substituted (cement and chemicals), while the use of biomass fuels combined with CCS can even yield negative emissions in strong decarbonisation scenarios.

## 5 Conclusions

As industry is a key contributor to GHG emissions and faces particular mitigation challenges compared to other energy de-mand sectors and the energy supply sector, it has to be considered both in detail and in conjunction with the entire system of

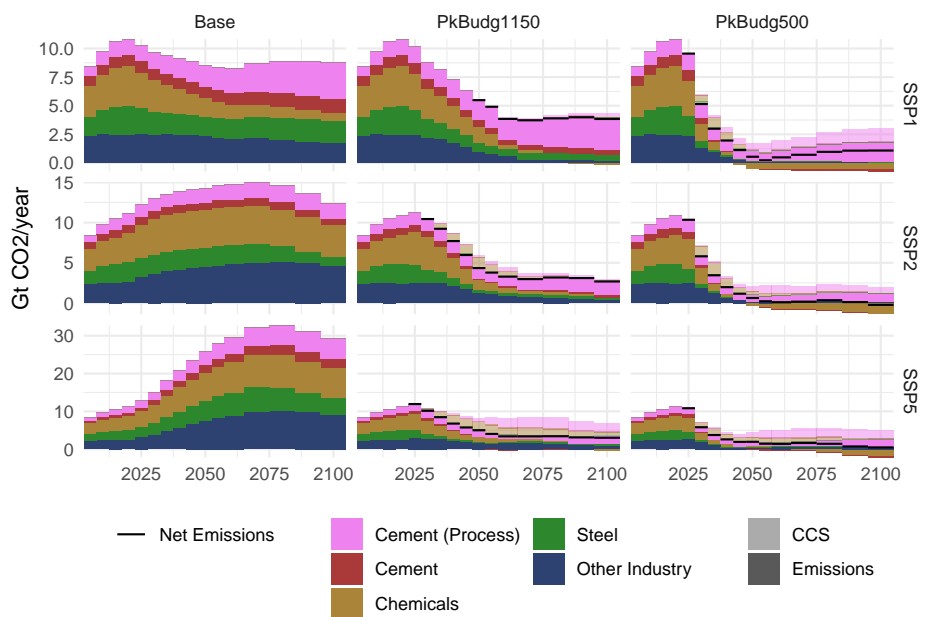

**Figure 13.** Industry CO$_2$ emissions and CCS. Net emissions are shown for periods with CCS or negative emissions (through bioenergy and CCS). Panels correspond to scenario matrix, with socioeconomic drivers (SSP1, SSP2, SSP5) and climate change mitigation policy (Base – no policy, PkBudg1150 and PkBudg500 – peak budgets of 1150 and 500 GtCO$_2$, corresponding to the 2°C and 1.5°C targets.)

macroeconomic production, energy demand, and carbon management options. The differentiation of the main industry subsectors enables the REMIND model to represent their different mitigation options and challenges in detail, while keeping them linked to the other economic sectors, the energy supply system, and the emissions, CCS, and CDR submodels, thus allowing for a consistent analysis of the entire energy-economy-climate system.

The projection of baseline scenarios of industry activity, energy demand, and CO$_2$ emissions is a prerequisite for developing and exploring climate change mitigation scenarios. The presented approach allows for both the generation of trajectories that replicate short-term trends and long-term global convergence of specific activity, as well as ways to conveniently and transparently modify those long-term trends to enable the creation of scenarios with different narratives to explore, like the different SSP scenarios.

The resulting REMIND model (in different stages of refinement) and scenarios were (e.g. Luderer et al. (2022)) and are currently (e.g. Schreyer et al. (submitted), Bauer et al. (in preparation)) used to investigate industry mitigation options and their implications for other economic sectors and the economy as a whole. A detailed analysis of industry subsector mitigation options and challenges that takes full advantage of these new modelling facilities is in preparation. It aims to spell out in detail the different drivers of emissions and their mitigation options, and to identify crucial bottlenecks for reducing the unavoidable

residual GHG emissions from industry that will have to be offset by CDR measures.

Possible future extensions include the integration of material flow analysis to improve the representation of the physical characteristics of different industry subsectors (e.g. an endogenous representation of in-use stocks of steel and plastics and their lifetimes as a limitation to their recycling, which are not captured well by the CES formulation), the introduction of feedbacks between other economic sectors and the demand for industry products (e.g. shrinking demand for steel through light-weighting of cars or the use of timber or carbon fibres instead of steel-reinforced concrete in construction), the effect of cheap variable electricity from renewables on direct and indirect (through hydrogen) electrification of industry, and international trade in energy intensive industrial goods and subsequent shift in production patterns of industry towards world regions with cheap abundant energy.

*Code availability.* The REMIND model is written in GAMS, with supporting scripts in R, and version 3.1.0 of the model is archived at Zenodo (https://doi.org/10.5281/zenodo.7628336).

The code for preparing REMIND input data is written in R and version 0.165.6 of the principal package mrremind is available at Zenodo (https://doi.org/10.5281/zenodo.10495588).

The source of this paper, including the R code for preparing the figures, is available at Zenodo (https://zenodo.org/doi/10.5281/zenodo.8272197). It is, however, not fully functioning as one proprietary data source (UNIDO INSTAT) cannot be shared publicly and had to be redacted.

*Data availability.* The data used for preparing REMIND input data is partly proprietary, and only avaliable upon request.

## Appendix A: REMIND Regions

The 21 region configuration for REMIND used to derive industry activity and final energy trajectories is comprised of regions with individual large countries (CHA – China, IND – India, JPN – Japan, USA – United States of America), country groups (CAZ – Canada, Australia, New Zealand, LAM – Latin America, MEA – Middle East and Northern Africa, OAS – other Asia, REF – former Soviet Union, SSA – Sub-Saharan Africa) and countries and country groups dividing Europe in roughly equal sized regions that are EU members (DEU, ECE, ECS, ENC, ESC, ESW, EWN, FRA) or not (NEN, NES, UKI; with Ireland being grouped with the United Kingdom even though the former is an EU member).

Tables A1 and A2 list the 21 regions and the countries grouped into them.

**Table A1.** Mapping of REMIND regions to countries

| Region | Countries |
|--------|-----------|
| CAZ | Australia; Canada; Heard Island and McDonald Islands; New Zealand; Saint Pierre and Miquelon |
| CHA | China; Hong Kong; Macao; Taiwan, Province of China |
| DEU | Germany |
| ECE | Czech Republic; Estonia; Latvia; Lithuania; Poland; Slovakia |
| ECS | Bulgaria; Croatia; Hungary; Romania; Slovenia |
| ENC | Aland Islands; Denmark; Faroe Islands; Finland; Sweden |
| ESC | Cyprus; Greece; Italy; Malta |
| ESW | Portugal; Spain |
| EWN | Austria; Belgium; Luxembourg; Netherlands |
| FRA | France |
| IND | India |
| JPN | Japan |
| LAM | Anguilla; Antarctica; Antigua and Barbuda; Argentina; Aruba; Bahamas; Barbados; Belize; Bermuda; Bolivia, Plurinational State of; Bonaire, Sint Eustatius and Saba; Bouvet Island; Brazil; Cayman Islands; Chile; Colombia; Costa Rica; Cuba; Curacao; Dominica; Dominican Republic; Ecuador; El Salvador; Falkland Islands (Malvinas); French Guiana; Grenada; Guadeloupe; Guatemala; Guyana; Haiti; Honduras; Jamaica; Martinique; Mexico; Montserrat; Nicaragua; Panama; Paraguay; Peru; Puerto Rico; Saint Barthelemy; Saint Kitts and Nevis; Saint Lucia; Saint Martin (French part); Saint Vincent and the Grenadines; Sint Maarten (Dutch part); South Georgia and the South Sandwich Islands; Suriname; Trinidad and Tobago; Turks and Caicos Islands; Uruguay; Venezuela, Bolivarian Republic of; Virgin Islands, British; Virgin Islands, U.S. |
| MEA | Afghanistan; Algeria; Bahrain; Egypt; Iran, Islamic Republic of; Iraq; Israel; Jordan; Kuwait; Lebanon; Libya; Morocco; Oman; Palestine, State of; Qatar; Saudi Arabia; Sudan; Syrian Arab Republic; Tunisia; United Arab Emirates; Western Sahara; Yemen |
| NEN | Greenland; Iceland; Liechtenstein; Norway; Svalbard and Jan Mayen; Switzerland |
| NES | Albania; Andorra; Bosnia and Herzegovina; Holy See (Vatican City State); Macedonia, the former Yugoslav Republic of; Monaco; Montenegro; San Marino; Serbia; Turkey |
| OAS | American Samoa; Bangladesh; Bhutan; British Indian Ocean Territory; Brunei Darussalam; Cambodia; Christmas Island; Cocos (Keeling) Islands; Cook Islands; Fiji; French Polynesia; French Southern Territories; Guam; Indonesia; Kiribati; Korea, Democratic People's Republic of; Korea, Republic of; Lao People's Democratic Republic; Malaysia; Maldives; Marshall Islands; Micronesia, Federated States of; Mongolia; Myanmar; Nauru; Nepal; New Caledonia; Niue; Norfolk Island; Northern Mariana Islands; Pakistan; Palau; Papua New Guinea; Philippines; Pitcairn; Samoa; Singapore; Solomon Islands; Sri Lanka; Thailand; Timor-Leste; Tokelau; Tonga; Tuvalu; United States Minor Outlying Islands; Vanuatu; Viet Nam; Wallis and Futuna |

**Table A2.** Mapping of REMIND regions to countries (continued)

| Region | Countries |
|---|---|
| REF | Armenia; Azerbaijan; Belarus; Georgia; Kazakhstan; Kyrgyzstan; Moldova, Republic of; Russian Federation; Tajikistan; Turkmenistan; Ukraine; Uzbekistan |
| SSA | Angola; Benin; Botswana; Burkina Faso; Burundi; Cameroon; Cape Verde; Central African Republic; Chad; Comoros; Congo; Congo, the Democratic Republic of the; Cote d Ivoire; Djibouti; Equatorial Guinea; Eritrea; Ethiopia; Gabon; Gambia; Ghana; Guinea; Guinea-Bissau; Kenya; Lesotho; Liberia; Madagascar; Malawi; Mali; Mauritania; Mauritius; Mayotte; Mozambique; Namibia; Niger; Nigeria; Reunion; Rwanda; Saint Helena, Ascension and Tristan da Cunha; Sao Tome and Principe; Senegal; Seychelles; Sierra Leone; Somalia; South Africa; South Sudan; Swaziland; Tanzania, United Republic of; Togo; Uganda; Zambia; Zimbabwe |
| UKI | Gibraltar; Guernsey; Ireland; Isle of Man; Jersey; United Kingdom |
| USA | United States of America |

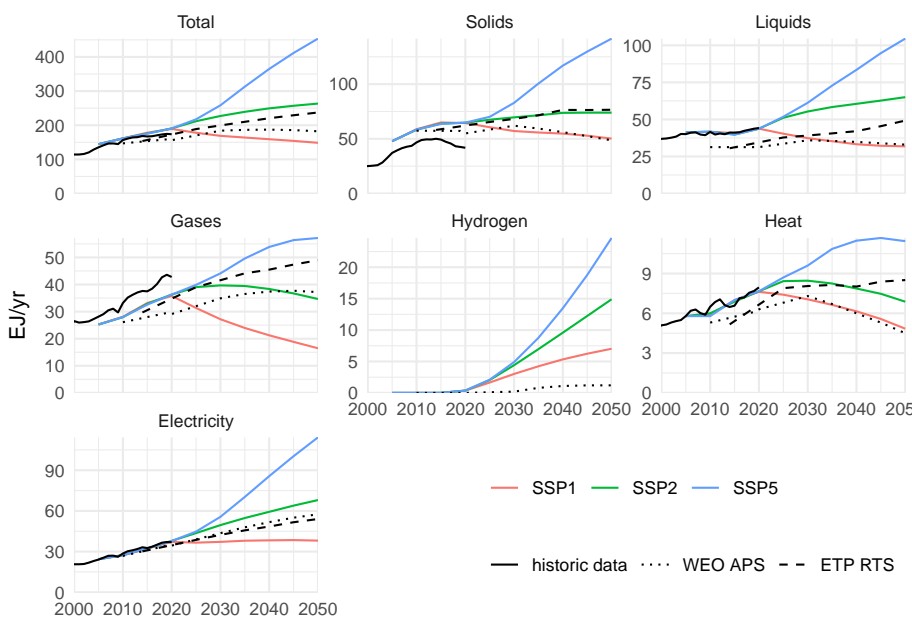

**Figure B1.** Comparison of industry final energy demand for the SSP1, SSP2, and SSP5 baseline scenarios to historic data (derived from International Energy Agency (2022)), the "Announced Pledges Scenario" (APS) from International Energy Agency (2021), and the "Reference Technology Scenario" (RTS) from International Energy Agency (2017).

## Appendix B:  Comparison to historic final energy demand

Figure B1 shows a comparison of industry final energy demand by energy carrier for the three baseline scenarios (SSP1, SSP2, and SSP5) with recent historic data (2000–2020, from International Energy Agency (2022)) as well as two examplary scenarios for near-term (until 2050) industry final energy demand, the "Announced Pledges Scenario" (APS, International Energy Agency (2021)), and the "Reference Technology Scenario" (RTS, International Energy Agency (2017)).

Note that all three sources (although stemming from the same source) differ in the way final energy carriers are aggregated and whether agriculture, fisheries, forestry, and refineries are included with industry or not.

*Author contributions.*  M.P. and G.L. designed the industry subsector formulation and scenario generation, which were implemented by M.P. The mark-up costs for hydrogen and high-temperature electricity were developed by F.S. The paper was written by M.P. with input from all authors. All work was supervised by G.L.

*Competing interests.*  The authors declare no competing interests.

*Acknowledgements.* The research leading to these results has received funding from the German Federal Ministry of Education and Research under grant agreements no. 03SFK5A (Ariadne) and the European Union's Horizon 2020 research and innovation programme under grant agreement No 101022622 (ECEMF).

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
