# Peer review of "Modelling Long-Term Industry Energy Demand and CO2 Emissions in the System Context Using REMIND (Version 3.1.0)"

_Geoscientific Model Development, 2023_

## Author Response (AR1)

**Author's Response**

We would like to thank both reviewers for their detailed and thoughtful comments. We will respond to them one-by-one, but rearranging them when they were addressed by the same change.

**Reviewer 1 comments**

> In general, paper addresses a relevant scientific modeling question within the scope of GMD and sufficiently advances modeling science by extending industry sector representation in a general equilibrium integrated assessment model REMIND which by default does not have the extended industry sector representation such as the ones added in this paper: cement, chemicals and iron&steel. It is an important model advancement that enables answering questions related to decarbonization of the hard-to-abate industry sectors within IAMs. Methods and assumptions that are explained are valid and clearly outlined, the overall structure is in general easy to follow with a clear language. The results are sufficient to serve as examples to understand the capabilities of the model.

Thank you for this assessment.
* * *
> The novelty and the original contribution of the model development is at the moment described but in different locations throughout (Section 2.5, Section 4) the paper and it is better to gather these in a dedicated part in the introduction section.

> Line 181–184: This section can be moved to introduction where the novelty of this model advancement is described.

We moved this paragraph, as suggested. (See also next comment.)
* * *
> Introduction explains the motivation of the model development well and supports with references why industry decarbonization is important in the context of climate change. Comparison to other IAMs and the current status of the modeling can be supported with more references apart from Bauer et al., which is not publicly available as mentioned in the general comments.

> In addition, the only reference used while explaining the novelty and while giving context about other Integrated Assessment Models is Bauer et al. which is not publicly available. In this case, the current status of the industry representation in IAMs can be supported with more references. As extending the industry sector representation in a general equilibrium model is the novel aspect of this paper, it would be essential to show references to some other IAMs and how they represent the industry sector, especially general equilibrium ones as they would be the closest to REMIND. With the relevant references, it can be emphasized if there are any other general equilibrium models that has similar industry extension or all of them represent industry as aggregated.

> Line 426-430, would be better to mention this in the introduction instead of in the middle of the exemplary results section. Also would be good to provide references for those other general equilibrium IAMs that are mentioned in the sentence that only have aggregated industry sector.

We moved this paragraph to the introduction, and extended it with a comparison of the pertinent features of other IAMs.

*There is a lack of integrated assessment modes representing price and economic feedbacks on industrial demand as well as efficiency, fuel switching and CCS options in industrial activities. In this respect, the extended industry sector in the REMIND model compares favourably to other IAMs. A forthcoming overview of industry $CO_2$ emission reduction (Bauer et al., in preparation) compares seven IAMs "with improved industry sector representation". They all represent at least the cement, chemicals, and steel production subsectors, and to varying degrees mitigation options like energy efficiency improvements, final energy substitution, and CCS. The endogenous reduction of industry demand as a trade-off with other mitigation options, is a feature of*

*general equilibrium models (two out of the seven: GEM-E3 (Fragkos and Fragkiadakis, 2022) and REMIND). Partial equilibrium models ((COFFEE: Rochedo (2016), MESSAGEix: Grubler et al. (2018), IMAGE: van Sluisveld et al. (2021), POLES: Després et al. (2018), PROMETHEUS: Fragkos et al. (2015)) rely on exogenous demand pathways for policy scenarios and are typically inelastic to changes in $CO_2$ and thus energy prices. The WITCH model (The WITCH team, 2017) is an example of another general equilibrium IAM, but only represents an aggregated stationary and not a separate industry sector.*
* * *
Line 57-59: It is mentioned new methods are used for producing consistent trajectories for industry subsector production and final energy use. It would be good to briefly explain what is new in these methods to give a hint to the reader on what to expect.

We extended the paragraph accordingly:

*Further to the refinements of industry sector representation within REMIND, we present new methods for producing consistent trajectories for industry subsector production and final energy use for different baseline scenarios used for calibrating the REMIND model. They project per-capita subsector activity based on per-capita GDP projections and allow for the variation of material and energy intensity to derive different scenarios.*
* * *
Is there a reference that explains the ESM (Energy System Model) section of the model more in details? Is it also part of the same formulation that maximizes intertemporal welfare? Does it determine which final energy carrier is used to satisfy final energy demand or does it just assign cost of production?

The Energy System Model of REMIND is explained in detail in Baumstark et al. (2021). We extended the paragraph to clarify that the ESM optimises quantities to be in equilibrium with the macroeconomic model.

*The integrated assessment model REMIND (Baumstark et al., 2021) is comprised of a macroeconomic growth model and a detailed energy system model (ESM), which are hard-linked, i.e. optimised simultaneously, with energy supply (by the ESM) and demand (by the macroeconomic model) quantities and prices in equilibrium.*
* * *
This section explains REMIND Model structure according to the title. At the end the description of the SSPs are given. I see the reason is to mention CES should be calibrated for different SSP baselines. That section (112-114) can be kept and the rest can be moved to where the description of SSPs is more relevant (e.g. Section 3 Projections and Input Data, 3.1 General Approach).

We moved these paragraphs to section 3.1
* * *
In line 147 the sentence starting with "electrification of clinker burning..." is not completely clear. It sounds like if you add electrification of clinker burning it is not possible to add CCS for process emissions which is almost half of the overall emissions and the other half would be the emissions from fuel combustion for the high-temperature heat. And electrification would be related to the emissions from combustion. Is it meant that electrification is still not well developed and costly so that it is not included and instead CCS is included as a mitigation option? Can be clarified.

We clarified this paragraph.

*This option is only included for the chemicals and other industry subsectors, since electric steel production is modelled explicitly (see section 3.3) and electrification of clinker burning (a major source of $CO_2$ emissions in cement production) is less attractive compared to CCS options, because it would only mitigate emissions from fuel burning, while large process emissions from limestone calcination (about half of total emissions) would remain (or require additional CCS).*
* * *
> In line 178, it could be better to explicitly mention which mitigation options are referred to. Demand for industrial goods (do you mean demand reduction?), industrial energy demand (do you mean energy efficiency measures and energy demand reduction?), energy production (fuel switching?).

We amended this paragraph.

*Different drivers and mitigation options are realised by different parts of the extended REMIND model as presented here. Population is an exogenous SSP scenario assumption and thus constant across scenarios (that are based on the same SSP). GDP and industrial activity are endogenous elements of the production function and vary with the strictness of mitigation constraints. Final energy, fossil fuels, carbon content and emissions are all endogenous elements of the ESM. Notably, the REMIND model covers the entire range of mitigation options, from reduced demand for industrial goods, over increased energy efficiency in industry, fuel-switching and renewable power production, to CCS.*
* * *
> The heading REMIND Implementation is confusing to me as I thought the previous sections 2.5, 2.4, 2.3 and 2.2 were already talking about REMIND implementation.

We expanded this to "Implementation of Mitigation Options in REMIND".
* * *
> Line 188: Which fuel is going to be used is determined in the macro-economic model based on elasticities and in ESM only the costs of producing that energy is determined?

We extended the relevant paragraph in section 2.1 (see answer to comment above) to clarify that the ESM and macroeconomic module are optimised simultaneously.
* * *
> Line 212 and 214: What is the logic behind 30% in 2050 and 8% in 2050 for hydrogen and electricity high temperature heat respectively?

We reformulated this and extended the following paragraphs to clarify that these values were chosen (like the mark-up costs) as a compromise between between baseline and policy scenario fidelity, with the goal of the marginal rate of substitution between hydrogen and HTH electricity being close to technical substitution levels in policy scenarios.

*First, we set future shares of hydrogen and high-temperature heat electricity for the baseline calibration. [...]*

*Second, we apply mark-up costs to both hydrogen and high-temperature heat electricity use in industry to represent additional cost of introducing new technologies to the production process that use these energy carriers.*

*Due to the economic nature of input substitution in the production function, the mark-up costs cannot be determined by techno-economic data and are instead set based on model behaviour. Both mechanisms, future baseline hydrogen/high-temperature heat electricity shares and mark-up costs, have been parametrised utilising the concept of the marginal rate of substitution, which describes the amount of one input needed to substitute another input to provide the same economic value (the ratio of the partial derivatives of two inputs into the production function). Final energy shares and mark-up cost are chosen such that the marginal rates of substitution with respect to gases and liquids (as the final energy carriers hydrogen and high-temperature heat electricity compete most strongly with) roughly approach technical substitution ratios in climate policy scenarios (one for hydrogen, two to three in for high-temperature electricity). The mark-up costs can be reduced in scenarios which e.g. stipulate a strong policy push for these technologies (Schreyer et al., submitted), making hydrogen and/or high- temperature heat electricity cheaper relative to other final energy carriers and in turn increasing their utilisation in industry.*
* * *
> Schreyer et al. not available publicly?

We extended the reference to clarify that Schreyer et al. has been submitted for peer-review.
* * *
Line 286 and 288: The sentence undermines the quality issues for the secondary steel production. It can be modified to acknowledge some of the limitations and increasing concerns about the recycled steel and the quality standards (https://pubs.acs.org/doi/10.1021/acs.est.7b00997).

We extended the paragraph to reflect current limitations to steel recycling as detailed in Daehn et al. (2017).

*Steel can mostly be collected, separated magnetically from other materials, and molten down to produce new products. (There are important constraints to the quality of recycled steel, due to tramp elements like copper (Daehn et al., 2017), currently limiting the share of recycled steel.) For these reasons, steel, unlike cement and chemicals, accumulates in the economy. Most iron ore that has been refined into steel stays available as steel.*
* * *
In general, in Section 3 different data sources are mentioned in between lines (United Nations Development Organization, Department of Economic and Social Affairs Population Division etc..) for different industry subsectors. It would be better to have this as a table of data sources for different subindustries instead of having them in between lines.

We added a table (Table 1) listing all data sources in section 3.1. We kept the data sources in the individual sections, too, as we feel it facilitates the understanding when reading up on one specific subsector.
* * *
In line 412, it would be good to explain and give reference what peak budget scenario is.

We extended the paragraph to clarify that the budgets are the cumulated emissions, and that they may decline after peaking due to net-negative emissions.

*The two mitigation scenarios use carbon price trajectories that result in cumulated GHG emissions (i.e. the carbon budget) from 2020 on to peak at 1150 and 500 Gt $CO_2$-equivalent ("PkBudg1150" and "PkBudg500") and subsequently decline through net-negative emissions, which are consistent with two possible interpretations of the climate targets of the Paris Accord (limiting global warming to 2°C with 67 % likelihood, or to 1.5°C with 50 % likelihood, cf. Arias et al. (2021), table TS.3).*
* * *
In line 486 [436], why comparing different years, 2070 for SSP2 and 2080 for SSP5?

We extended the sentence to clarify that SSP2 and SSP5 FE demand peaks in in 2070 and 2080, respectively, and declines afterwards, where as SSP1 FE demand declines immediately.

*The SSP1 Base scenario shows an immediate decline in industry FE demand, in accordance with the scenario narrative of sustainability and efficiency, while the SSP2 and SSP5 scenarios display an increase and subsequent decrease in FE demand, albeit to markedly different levels (273 EJ/year peak in 2070 for SSP2, and 559 EJ/year peak in 2080 for SSP5).*
* * *
In line 437, share of electricity in which year?

We extended the paragraph to include that the maximum electricity shares for PkBudg1150 and PkBudg500 are reached in 2055 and 2080, respectively.

*The share of electricity (an indicator of industry decarbonisation) rises to 28 % in 2090 for the Base scenarios, and 45–49 % in 2080 (PkBudg1150) and 2055 (PkBudg500) for the mitigation scenarios, while the share of solids decreases strongly, to be partially replaced with liquid final energy carriers, part of which will be generated from biomass.*
* * *
In the explanation of Figure 9, the changes of the industry activity are described as "higher", "lower", "reduction in industry demand". It would be good to quantify these with numbers in the text to have a better idea on the magnitude at least for 2050 and/or 2100. Similarly, for shift from primary to secondary steel.

We extended the paragraphs and added the 2015 to 2100 percentage changes.

*In accordance with the SSP scenario narratives (O'Neill et al., 2017, Bauer et al. (2017)) (see section 3.7), the SSP1 Base scenario shows lower growth in industry demand (-30 to 210 %, 2015 to 2100 change) than the SSP2 scenario (-47 to 360 %), while the SSP5 Base scenario shows higher growth (140 to 810 %), reproducing the data from the projection and scenario variation steps (section 3). The "other industry" subsector sees the highest growth across all socioeconomic baselines (210 to 810 %), in line with the assumed increase in the share of industries producing high-value goods as opposed to bulk materials. The increase in the chemicals subsector's activity (46 to 690 %), too, agrees with the assumption of higher shares of products from light metals like aluminium, plastics and other modern materials, rather than steel.*
* * *
I am curious about the role of hydrogen in the steel sector in these scenarios. I see certain share in the final energy mix but it is not mentioned in the text. It is only mentioned steel sector moves from primary to secondary production drastically cutting the use of coal. In 1.5 degree scenario steel sector has no more emissions already before 2050. Is all the emissions reduction in steel sector attributed to recycling which doesn't sound realistic? Is there CCS in steel industry? These can be worthwhile to mention in the explanation of the reduction of emissions from steel industry.

The paragraph is intended to highlight the differences in mitigation options employed in the different subsectors, and not to give a full account of subsector decarbonisation, which is the subject of a forthcoming paper currently in preparation (cf. section 5). We extended the paragraph to give a fuller picture of subsector decarbonisation, especially for steel.

*The steel sector decarbonises by expanding production of secondary steel that uses carbon-free electricity (within the limits of scrap availability), by reducing production of primary steel, and by switching to carbon-free fuels. The other industry subsector substitutes electricity for fuels, while also increasing energy efficiency, so the total electricity demand increases only slightly, and moves to carbon-free fuels. Notably, the residual share of fossil fuels is lowest in the other industry subsector, as it does not allow for the application of CCS.*
* * *
Finally, it would be good to see somewhere (in supplementary) description of model regions.

We added a description of model regions, and a table listing all countries belonging to the regions, in Appendix A.
* * *
In line 467, 'The resulting REMIND model (in different versions) and scenarios...". It was not clear what is meant by different versions.

We extended the paragraph to clarify that the improved REMIND industry representation was already and is currently used, while work on it continues.

*The resulting REMIND model (in different stages of refinement) and scenarios were (e.g. Luderer et al. (2022)) and are currently (e.g. Schreyer et al. (submitted), Bauer et al. (in preparation)) used to investigate industry mitigation options and their implications for other economic sectors and the economy as a whole.*
* * *
In conclusion, the limitations of the model can also be discussed next to the advantages and the novelty introduced. There are a couple of words about improving physical characteristics of different industry subsectors, this limitation and others can be explained a bit more in detail and then can be connected to the possible future extensions.

We extended the description of future work.

*Possible future extensions include the integration of material flow analysis to improve the representation of the physical characteristics of different industry subsectors (e.g. an endogenous representation of in-use stocks of steel and plastics and their lifetimes as a limitation to their recycling, which are not captured well by the CES formulation), the introduction of feedbacks between other economic sectors and the demand for industry products (e.g. shrinking demand for steel through light-weighting of cars or the use of timber or carbon fibres instead of steel-reinforced concrete in construction), the effect of cheap variable electricity from renewables on direct and indirect (through hydrogen) electrification of industry, and international trade in energy intensive industrial goods and subsequent shift in production patterns of industry towards world regions with cheap abundant energy.*

**Reviewer 2 comments**

In general, this paper shows the author's in-depth understanding of industrial energy use. Based on the judgment of the energy use characteristics of different industry subsectors, reasonable model classification and carbon reduction approaches are set, which better realizes the extension of industry modeling within the REMIND integrated assessment model to industry subsectors. But at the same time, this paper still has some specific problems that need further supplementation and improvement:

Thank you for this assessment and the valuable suggestions.
* * *
The introduction part needs to be supplemented with a more systematic literature review to explain in detail the current status and existing shortcomings of the IAM model. Is there any researches done similar work to develop subdivision modules for the industrial sector? Reflecting the innovativeness of the model in this study is core and critical.

We extended the introduction it with a comparison of the pertinent features of other IAMs.

*There is a lack of integrated assessment modes representing price and economic feedbacks on industrial demand as well as efficiency, fuel switching and CCS options in industrial activities. In this respect, the extended industry sector in the REMIND model compares favourably to other IAMs. A forthcoming overview of industry $CO_2$ emission reduction (Bauer et al., in preparation) compares seven IAMs "with improved industry sector representation". They all represent at least the cement, chemicals, and steel production subsectors, and to varying degrees mitigation options like energy efficiency improvements, final energy substitution, and CCS. The endogenous reduction of industry demand as a trade-off with other mitigation options, is a feature of general equilibrium models (two out of the seven: GEM-E3 (Fragkos and Fragkiadakis, 2022) and REMIND). Partial equilibrium models ((COFFEE: Rochedo (2016), MESSAGEix: Grubler et al. (2018), IMAGE: van Sluisveld et al. (2021), POLES: Després et al. (2018), PROMETHEUS: Fragkos et al. (2015)) rely on exogenous demand pathways for policy scenarios and are typically inelastic to changes in $CO_2$ and thus energy prices. The WITCH model (The WITCH team, 2017) is an example of another general equilibrium IAM, but only represents an aggregated stationary and not a separate industry sector.*
* * *
Section 2.2, the impact of replacing physical measure with monetary measure needs to be considered, or at least discussed. The monetary value of the same products produced in different countries vary greatly, and there is different subdivided product structures in different country, which will result in differences between monetary and physical measure.

We extended the section by a paragraph discussing the impact of using value added instead of monetary production.

*Using value added instead of physical production to drive industry energy demand incurs two difficulties. Both the specific value added per unit of (physical) production and the composition of different types of products making up subsector production vary across regions and change over time. This reduces the interpretability of*

*subsector production figures given in value added, especially in absolute terms. It does, however, not impinge on the usefulness for linking economic activity and industry energy demand, as the historical regional differences are subsumed by the regression of subsector energy demand on subsector activity, and the composition of subsector production is expected to move in the direction of higher shares of high-value products (decreasing physical production per unit value added) as economies evolve to higher GDP per capita, which acts in the same direction of increasing energy efficiency (decreasing energy demand per unit physical production), not introducing behaviour that is different from subsectors with physical representation.*
* * *
Why are fossil fuels divided into solids, liquids, and gases instead of coal, oil, and gas? What are the advantages of the former?

This follows common practice in integrated assessment modelling, as not all solids, liquids, or gases are fossil-based: Final energy carriers (used in demand sectors) are separated according to their use characteristics (solids, liquids, and gases are transported, stored, and burned differently), while their provisioning (from fossil or biogenic sources, or based on hydrogen) is part of the energy supply sector and does not impinge on their utility (within limits: natural gas and biogas are quasi-perfect substitutes, charcoal can be used in stead of coke in steelmaking, but requires process adaptations).

We also extended the caption of figure 10 to clarify this point.

Solids, Liquids, and Gases combine final energy carriers from fossil and non-fossil sources.
* * *
Line 211-216, please explain the basis for these value settings.

We reformulated this and extended the following paragraphs to clarify that these values were chosen (like the mark-up costs) as a compromise between between baseline and policy scenario fidelity, with the goal of the marginal rate of substitution between hydrogen and HTH electricity being close to technical substitution levels in policy scenarios.

*First, we set future shares of hydrogen and high-temperature heat electricity for the baseline calibration. [. . .]*

*Second, we apply mark-up costs to both hydrogen and high-temperature heat electricity use in industry to represent additional cost of introducing new technologies to the production process that use these energy carriers.*

*Due to the economic nature of input substitution in the production function, the mark-up costs cannot be determined by techno-economic data and are instead set based on model behaviour. Both mechanisms, future baseline hydrogen/high-temperature heat electricity shares and mark-up costs, have been parametrised utilising the concept of the marginal rate of substitution, which describes the amount of one input needed to substitute another input to provide the same economic value (the ratio of the partial derivatives of two inputs into the production function). Final energy shares and mark-up cost are chosen such that the marginal rates of substitution with respect to gases and liquids (as the final energy carriers hydrogen and high-temperature heat electricity compete most strongly with) roughly approach technical substitution ratios in climate policy scenarios (one for hydrogen, two to three in for high-temperature electricity). The mark-up costs can be reduced in scenarios which e.g. stipulate a strong policy push for these technologies (Schreyer et al., submitted), making hydrogen and/or high- temperature heat electricity cheaper relative to other final energy carriers and in turn increasing their utilisation in industry.*
* * *
line 295-297, 302-303, 334-335 etc. These descriptions of per capita GDP and population data sources should be integrated and put in the general approach.

We added a table (Table 1) listing all data sources in section 3.1. We kept the data sources in the individual sections, too, as we feel it facilitates the understanding when reading up on one specific subsector.
* * *
figure 3 Please confirm whether the ordinate axis corresponds to the unit.

The scaling of the ordinate was wrong and has been corrected.
* * *
line 336-337 figure 6 and 7, there is no regression function in the figures,

We harmonised the display of regression functions between figures 2–7, emphasising the regression functions in figures 6 and 7 (red lines instead of dashed black lines).

how is the regression function selected?

This is addressed in section 3.1, as the regression function is identical for all subsectors.

*This formulation is used because it presupposes a decoupling of per-capita demand from increasing affluence levels, and its positive codomain makes it easily tractable. It was found to be a good fit to historic data in previous research (van Ruijven et al., 2016).*
* * *
It is best to add verification of the model results in this article, at least for the base year, to increase the credibility of the model.

We added Appendix B, comparing the SSP1, SSP2, and SSP5 baseline projections to historic data and two scenarios of industry final energy demand by the IEA.

---

## Author Response (AR2)

**Author's Response**

Thank you for adding a basic evaluation of energy demand relative to historic data in Appendix B. Since both reviewers brought this up, it would be useful to highlight, so please refer to Appendix B—and briefly summarize the results—somewhere in the main text.

We added a reference to Appendix B in section 4:

*For a comparison of total industry FE demand with historic data and other projections, see Appendix B. Total FE demand of the SSP2 scenario increases in accordance with the "Reference Technology Scenario" from International Energy Agency (2017) until 2050 (cf. section 3.6), with SSP1 and SSP5 markedly higher and lower, respectively.*

- L58: "modes" should be "models"
- L189: "recent" should be "forthcoming" or something similar
- Table 1: Extra "steel" in third row first column?
- L518: "Irland" typo
- L519: Replace "it" with "the former"
- Data availability statement: "propriatory" should be "proprietary"
- L530: "supervisedlibrar"

We addressed all these typos.

Font is too small in Figs. 2-8

We increased the font size in the figures.